# GRAPH-BASED NEAREST NEIGHBORS WITH DYNAMIC UPDATES VIA RANDOM WALK

**Nina Mishra**
Amazon
nmishra@amazon.com

**Yonatan Naamad**
Amazon
ynaamad@amazon.com

**Tal Wagner**
Amazon and Tel Aviv University
tal.wagner@gmail.com

**Lichen Zhang**
MIT CSAIL
lichenz@csail.mit.edu

## ABSTRACT

Approximate nearest neighbor search (ANN) is a common way to retrieve relevant search results, especially now in the context of large language models and retrieval augmented generation. One of the most widely used algorithms for ANN is based on constructing a multi-layer graph over the dataset, called the Hierarchical Navigable Small World (HNSW). While this algorithm supports insertion of new data, it does not support deletion of existing data. Moreover, deletion algorithms described by prior work come at the cost of increased query latency, decreased recall, or prolonged deletion time. In this paper, we propose a new theoretical framework for graph-based ANN based on random walks. We then utilize this framework to analyze a randomized deletion approach that preserves hitting time statistics compared to the graph before deleting the point. We then turn this theoretical framework into a *deterministic* deletion algorithm, and show that it provides better tradeoff between query latency, recall, deletion time, and memory usage through an extensive collection of experiments.

## 1 INTRODUCTION

We study the *approximate nearest neighbor search problem* (ANN): given a collection of $n$ points $P \subset \mathbb{R}^d$, a parameter $k$ and a query point $q \in \mathbb{R}^d$, the goal is to find the top-$k$ points in $P$ that are closest to $q$ under some distance measurements such as $\ell_2$ distance or cosine similarity[1]. This problem is a fundamental component of Retrieval Augmented Generation (RAG), widely adopted to improve the accuracy of Large Language Models (LLM) (Lewis et al., 2020; Gao et al., 2023; Jiang et al., 2023; Fan et al., 2024). One of the most popular ways to solve the ANN problem is through graph-based approaches, where a data-dependent graph is constructed of the dataset, and the queries can be quickly routed on such graph (Jayaram Subramanya et al., 2019; Malkov & Yashunin, 2020; Fu et al., 2019). In this work, we focus on the Hierarchical Navigable Small World (HNSW) graph (Malkov & Yashunin, 2020), a data structure with strong practical performances. This is a multi-layer graph data structure, where the higher layers of the graph tend to have fewer vertices and "long-range" edges that help to navigate between clusters, while the lower layers have "short-range" edges that enhance the connectivity within the clusters. To perform a query search, one starts at the higher layers via the long-range edges to make coarse-grained progress towards the clusters containing the nearest neighbors, and drops to lower layers for finer-grained progress to find the nearest neighbors within the cluster.

While the HNSW algorithm can naturally handle insertions (Harwood et al., 2024), they do not possess *deletion* capabilities. This becomes more and more problematic as modern datasets are highly dynamic and require many deletions. For example, advertisers may remove ads when the campaign budget is exhausted; a clothing retailer may remove spring clothing in the fall and remove fall clothing in the spring; a shoe company may eliminate a previous shoe version when a new generation drops. Thus, deletion is a common and necessary operation.

---

[1]In this paper, we use $\| \cdot \|$ to denote such abstract distance measurement.

Since the HNSW algorithm does not come with a deletion procedure, a common, practical approach is to never delete a point at all. Instead, the corresponding vertex in the graph is marked with a "tombstone" indicating that the point should never be outputted as a nearest neighbor. At query time, these tombstones can either be handled in post-processing (the corresponding points would be removed from the output) or in real-time (modifying the HNSW algorithm to never stop its search at a tombstone). We consider this algorithm as a baseline for our proposed approach. The advantage of this approach is that graph navigability is preserved, and therefore tombstones maintain high recall. However, as tombstones are never removed from the graph, the query latency rises significantly when there are large amounts of tombstones, as the search would take many more steps to reach the desired nearest neighbors. Moreover, the memory usage of the data structure stays constant while the number of points declines, leading to unnecessary storage.

Due to the importance of developing a good deletion algorithm with improved query throughput and memory usage, a rich literature has considered approaches beyond tombstones. An even simpler idea is to delete the point from the graph and the data structure without patching the graph (we will refer to it as a 'no patching' algorithm). Such an approach suffers when there are many deletions or queries are drifting towards the clusters being deleted, resulting in much worse recall (Xu et al., 2022). Another class of deletion algorithms is to reconnect the subgraph after deletion as in (Singh et al., 2021; Xu et al., 2022; Xiao et al., 2024; Zhao et al., 2023; Xu et al., 2023; 2025). The reconnection strategy ranges from local to global: if we want to delete $p$, a local reconnect attempts to find for every $u \in N(p)$, the neighborhood of $p$, another point $v \in N(p)$ that is the nearest neighbor of $u$. This approach has low query latency and slightly improved recall compared to no patching, but the recall is still significantly lower than in other alternatives. To further improve recall, Singh et al. (2021) proposes the FreshDiskANN algorithm that uses a 2-hop neighborhood: instead of only rerouting within $N(p)$, FreshDiskANN adds edges between $u$ and $N(p) \cup N(u)$, then prunes edges to ensure the sparsity of the graph. A more global approach introduced by Xu et al. (2022) (where they termed the algorithm global reconnect) is to re-insert all the points in $N(p)$ and improve the connectivity of the subgraph by utilizing the robustness of the HNSW insertion procedure. Both FreshDiskANN and global reconnect suffer from much longer deletion times due to the non-local nature of the algorithm and the need to interact with larger subgraphs, and require nontrivial effort to parallelize the insertions.

**Our results.** Our main contribution is a deletion procedure that arises through a theoretically grounded twin framing of the HNSW algorithm. This algorithm enjoys good recall, query speed, overall deletion time and memory usage, as summarized in Table 1. In the original HNSW algorithm, for a given query $q$, the algorithm walks to the adjacent vertex $u$ that minimizes $\|q - u\|$, in a deterministic and greedy fashion. In our twin formulation, we "soften" it by walking to a random neighbor with probability proportional to $\exp(-r^2 \cdot \|q - u\|^2)$ for $r > 0$. We call this a "softmax walk". While highly similar to the "hard" greedy walk (as we confirm experimentally), this random walk interpretation leads to a theoretical deletion procedure that maintains the properties of the original random walk. Specifically, the algorithm functions by first computing local edge weights over $N(p)$ that precisely preserve the random walk probability, then using a simple randomized sparsification scheme that approximates the *hitting time* of a walk, i.e., the expected number of steps it takes for a random walk at a vertex $u$ to reach a vertex $v$. While this does not directly guarantee that we will hit the same nearest neighbors exactly, it does ensure that we make a similar number of steps to the target as before. We then turn this randomized, theoretical algorithm into a deterministic, practical algorithm: instead of random sampling edges, we could compute the edge weights first then simply take heaviest edges. We conduct extensive experiments on various datasets in the *mass deletion setting*, where large fraction of the points are removed from the dataset and show that our method has strong advantage when points are frequently deleted in large batches. We show the edge of our method over other alternatives in four key metrics: recall, query speed, deletion time and memory usage.

## 2 RELATED WORK

**Approximate Nearest Neighbor Search.** Approximate nearest neighbor (ANN) search is a core algorithmic task with a long line of research, both in theory and in practice. Theoretically, Locality Sensitive Hashing (LSH) (Indyk & Motwani, 1998; Andoni & Indyk, 2008; Andoni & Razenshteyn, 2015; Andoni et al., 2015; 2017; 2018; Dong et al., 2020) has been a popular solution with provably

Table 1: Summary of HNSW deletion algorithms.

| Method | Recall | Speed | Del Time | Space |
|---|---|---|---|---|
| Tombstone | ✓ | ✗ | ✓ | ✗ |
| No patch | ✗ | ✓ | ✓ | ✓ |
| Local | ✗ | ✓ | ✓ | ✓ |
| FreshDiskANN | ✓ | ✓ | ✗ | ✓ |
| Global | ✓ | ✓ | ✗ | ✓ |
| SPatch (ours) | ✓ | ✓ | ✓ | ✓ |

fast query time and efficient memory usage. While LSH offers good performance in theory, its practical variants are usually *data-oblivious*. On the other hand, practical ANN algorithms are oftentimes *data-dependent*; this class of algorithms includes quantization methods (Jégou et al., 2010; Ge et al., 2013; Kalantidis & Avrithis, 2014; Yue et al., 2024; Gao & Long, 2024) and graph-based methods (Malkov & Yashunin, 2020; Jayaram Subramanya et al., 2019; Wang et al., 2021; Fu et al., 2019; Harwood & Drummond, 2016; Groh et al., 2023), the latter of which are the focus of this work. In these algorithms, the points in the dataset are coalesced into a specially-constructed proximity graph. To handle a query, one runs a greedy search algorithm to traverse the graph (i.e., iteratively move from the current point to its adjacent point closest to the query vector) to find the approximate nearest neighbors of the query. Among this body of work, we focus on the Hierarchical Navigable Small World (HNSW) algorithm of Malkov & Yashunin (2020), due to its wide adoption, fast query throughput, and good recall characteristics. A key limitation of HNSW is that it only explicitly supports insert and query operations, and implicitly assumes that the dataset is either static or only incremental. In practice, deletions (if any) are usually handled ad-hoc; the industry standard is to mark any deleted vectors as unreturnable "tombstones", and periodically rebuild the data structure entirely from scratch at great cost (Xu et al., 2022). Some more recent approaches try to directly incorporate some notion of deletion into the graph data structure, preempting the need for periodic batch rebuilds (Singh et al., 2021; Xu et al., 2022; Xiao et al., 2024; Xu et al., 2023; Zhao et al., 2023; Xu et al., 2025). These methods typically operate by first excising the deleted node from the graph, then "patching" the graph by adding new connections among the removed node's former neighborhood. However, most of these approaches offer no theoretical guarantees and typically come with both performance and recall penalties on real-world datasets, especially for mass deletion.

**Graph Sparsification and Random Walks.** Given a graph, an elementary way to explore it is through random walks, which is both practical (Hamilton et al., 2017) and has rich theoretical connections to electrical networks (Doyle & Snell, 1984; Tetali, 1991) and spectral graph theory (Chung, 1997; Spielman, 2007). Important statistics of random walks, including hitting time and commuting time (Aldous & Fill, 2002) can be computed by solving linear systems in the graph Laplacian matrix (Merris, 1994). A popular approach to improve the efficiency of solving Laplacian linear systems is via *spectral sparsification* that reduces the number of edges in the graph while preserving all Laplacian quadratic forms by sampling according to the effective resistances of edges (Spielman & Srivastava, 2011; Batson et al., 2009). All state-of-the-art solvers for Laplacian systems utilize spectral sparsification (Spielman & Teng, 2004; Koutis et al., 2010; Peng & Spielman, 2014; Cohen et al., 2014; Kyng & Sachdeva, 2016). In this work, we focus on another type of sparsification, based on sampling by row norms (Drineas & Kannan, 2001; Frieze et al., 2004; Kannan & Vempala, 2017). It is conceptually simpler and more efficient to implement, though it provides weaker guarantees.

## 3 PRELIMINARIES

**HNSW.** The Hierarchical Navigable Small World (HNSW) data structure proposed in Malkov & Yashunin (2020) is a graph-based ANN data structure that utilizes a *hierarchical* structure. In particular, it is a sequence of undirected graphs[2] with each graph in the sequence called a "layer". For the reminder of this paper, we assume there are $L$ layers of graphs, with the top one as the $L$-th layer and the bottom one as the 1st layer. The bottom layer of the HNSW contains one vertex for each point in the dataset, and each other layer contains a random subset of points in the layer below. In addition to internal edges within a layer, the two vertices representing the same point in two consecutive layers also have a vertical edge between them so that a search could traverse between layers.

---

[2]In some library implementations such as FAISS, directed graphs are used instead.

The fundamental operation of the HNSW is the search operation. Given a query point $q$, the search starts at the $L$-th layer with an entry point. At any timestamp $t$, we let $u_t$ be the point where the query $q$ is currently on with $u_0$ being the entry point, then we move $q$ to $u_{t+1} := \arg\min_{v \in N(u_t)} \|q - v\|$. When the search can no longer make progress, it uses the vertical edge to move one layer down, and repeat the process until the bottom layer. In the reminder of this paper, we refer to this procedure as the "greedy search". The insertion is then executed by running greedy search on the point-to-be-inserted (with more entry points to move down layers) and add edges along the way. For a more comprehensive overview of HNSW and related algorithms, see Appendix D.

HNSW and other graph-based nearest neighbor search data structures have also been studied through the lens of theory. Laarhoven (2018) studies the performance of greedy search when the size of the dataset $n = 2^d$. Fu et al. (2019) analyzes the time and space complexity of searching in the monotonic graphs, and it requires the query to be one of the points in the dataset. Prokhorenkova & Shekhovtsov (2020) proves that the greedy search can be done in sublinear time on the plain nearest neighbor graph in both the dense and sparse regime, and adding vertical edges as in HNSW effectively reduces the number of steps to find the correct nearest neighbor. Shrivastava et al. (2023) relaxes the assumption of nearest neighbor graphs to approximate nearest neighbor graphs. Indyk & Xu (2023) crafts a condition called $\alpha$-shortcut reachability, and proves a class of graph-based algorithms are provably efficient for $\alpha$-shortcut reachable graphs. Lu et al. (2024) further speeds up the search process by using approximate distances instead of exact distances and proves probabilistic guarantees for this approximation. Diwan et al. (2024) considers a class of simplified HNSW graphs and proves the navigability under certain construction algorithm. Oguri & Matsui (2024) shows that adaptively choosing the entry point provably functions better than using a fixed entry point.

**Deletion Strategies for Graph-based ANN.** One could alternatively interpret HNSW as a multi-layer version of the DiskANN data structure (Jayaram Subramanya et al., 2019), with improved navigability between distant clusters. While these data structures are naturally attuned for insertions, deletion is much more challenging and is heavily based on heuristics. What would be some metrics we'd want a good deletion algorithm to have? 1). Memory usage. We would like the space consumed by the data structure to be proportional to the number of points stored in the data structure; 2). Efficiency. We would like the deletion algorithm to be performed efficiently, and ideally, as efficient as the search algorithm; 3). Recall. The deletion algorithm should not impair the recall performance of the algorithm. If one only cares about the recall, a theoretically "optimal" strategy could be developed:

**Theorem 3.1.** *Let $P \subset \mathbb{R}^d$ be an $n$-point dataset preprocessed by an HNSW and $p \in P$ be a point to-be-deleted that is not the entry point. Fix a query point $q \in \mathbb{R}^d$ and suppose the search reaches layer $l \in \{1, \ldots, L\}$, let $N(p)$ denote the neighborhood of $p$ at layer $l$. Suppose $q$ reaches $N(p)$, visits and leaves $p$. Consider the deletion procedure that removes $p$ at layer $l$ and forms a clique over $N(p)$, then the search of $q$ on the new graph is equivalent to the search of $q$ on the old graph.*

We defer the proof to Appendix C. The above theorem indicates that it is enough to design a data structure that emulates the subgraph without deleting $p$. Two obvious choices for an exact data structures are 1). Tombstone, where the subgraph structure is preserved and we will still visit $p$; 2). Clique, where all possible paths in $N(p)$ are preserved.

These two approaches share similar drawbacks. While the tombstone algorithm cannot ever free memory and reduces query throughput through lengthened walks, the clique algorithm densifies subgraphs and reduces throughput by increasing the number of distance calculations made per step.

## 4 SPATCH: VERTEX DELETION VIA RANDOM WALK PRESERVING SPARSIFICATION

To motivate our deletion algorithm, we develop a theoretical framework for analyzing HNSW that, *instead of walking to the nearest point over the neighborhood, performs a random walk with probability given by the softmax of the squared distance.*

Specifically, let $q$ be the query point and suppose the search is currently on point $u$. Instead of deterministically moving to $\arg\min_{c \in N(u)} \|c - q\|$, we move to a random $c \in N(u)$ with probability $\frac{\exp(-r^2 \cdot \|c-q\|^2)}{\sum_{v \in N(u)} \exp(-r^2 \cdot \|v-q\|^2)}$. We call this search algorithm the ***softmax walk***. While this is different from greedy search, we empirically validate that the performance of the softmax walk for large

enough $r$ is very similar to that of the greedy walk (see Section 5). This simple modification enables us to analyze the graph search from the perspective of a random walk. Instead of interpreting the HNSW graph as an unweighted graph, we can now view it as a *weighted* graph with edge weight determined by the query point $q$: letting $u$ be the point in the graph that the search for $q$ is currently on, for any edge $\{u, v\}$, the edge weight is $w(u, v) = \exp(-r^2 \cdot \|q - v\|^2)$. The HNSW search then performs a random walk on this weighted graph. We could interpret it as a probabilistic HNSW:

- Instead of greedy search, we use *random walk*;
- Instead of taking the top-$t$ nearest neighbors, we *sample $t$ edges proportional to edge weights*.

The edge weights we choose are equal to the Gaussian kernel between the query and the dataset. Intuitively, this is particularly useful to emulate greedy search because it helps differentiate the distances: if $q$ is far away from $v$, then $\exp(-r^2 \cdot \|q - v\|^2)$ would be much smaller than a $v'$ that is closer to $q$. Hence, under a proper choice of $r$, the Gaussian kernel function assigns exponentially high probability to nearby points, emulating the greedy search.

We remark that graphs with Gaussian kernel weights are very widely used. Note, however, the difference in our case: the weight of an edge $\{u, v\}$ is not $\exp(-r^2 \cdot \|u - v\|^2)$ as it would normally be, but rather, the edge is viewed as two directed edges $u \to v$ and $v \to u$, with respective weights $\exp(-r^2 \cdot \|q - v\|^2)$ and $\exp(-r^2 \cdot \|q - u\|^2)$. Thus, the edge weight is independent of its source point, and varies by the query $q$ being searched.

Thus, the graph constructed by our twin HNSW formulation can be formally viewed as the result of randomized graph sparsification. This algorithm can be integrated into the multi-layer HNSW structure by repeating the procedure from layer $l$ to 0. This weighted graph sparsification perspective provides the motivation and theoretical foundation of our deletion algorithm.

As discussed in Section 3, we would like a deletion algorithm that is optimized for memory usage, efficiency, and recall. Since we are now working with random walks, we first prove a variant of Theorem 3.1 in our model: instead of preserving the exact search path after deletion, we aim to preserve the random walk probabilities after deletion.

**Theorem 4.1.** *Let $G = (V, E, w)$ be a weighted graph. Define the random walk under the weights $w$ for any edge $\{u, v\} \in E$ as $\Pr[u \to v \mid w] = \frac{w(u,v)}{\deg(u)}$ where $\deg(u) = \sum_{z \in N(u)} w(u, z)$. Let $p$ be the point to be deleted. For all $u, v \in N(p)$, define the new weights $w'(u, v)$ as $w'(u, v) = w(u, v) + \frac{w(u,p) \cdot w(p,v)}{\deg(p)}$. Let $E(p) = \{\{u, p\} : u \in N(p)\}$ and $C(p) = \{\{u, v\} : u \neq v, u, v \in N(p)\}$. Then for the new graph $G' = (V \setminus \{p\}, E \setminus E(p) \cup C(p), w')$, we have*

$$\Pr[u \to v \mid w'] = \Pr[(u \to p \to v) \vee (u \to v) \mid w].$$

The proof is deferred to Appendix C. This theorem is known as a "star-mesh transform", and arises from Schur complements and Gaussian elimination (Rosen, 1924; Dorfler & Bullo, 2012; Wagner et al., 2018). The new edges $C(p)$ induce a clique over the neighborhood of $p$, making the graph denser than before. To sparsify the graph, we adapt a simple strategy: sampling edges according to edge weights. The complete algorithm, called `SPatch`: Sparsified Patching, is given in Algorithm 1 and shown in Figure 1.

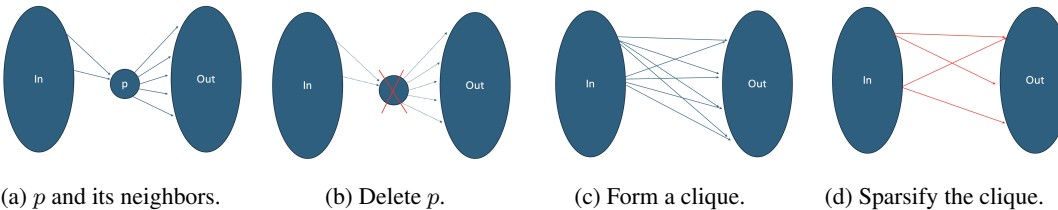

(a) $p$ and its neighbors.    (b) Delete $p$.    (c) Form a clique.    (d) Sparsify the clique.

Figure 1: The deletion procedure of Algorithm 1. It proceeds by first forming a clique over the neighborhood of a deleted point, and then sparsifies the clique according to edge weights.

To prove theoretical guarantees for Algorithm 1, we need to introduce a few linear algebraic definitions related to graphs.

---

**Algorithm 1** SPatch: Sparsified Patching

---

1: **procedure** SPATCH($G(V, E); p \in V; r, \alpha, t \geq 0$)
2:                                                                  $\triangleright$ $p$ is the point to be deleted from the graph.
3:     Define $w(u, v) = \exp(-r^2 \cdot \|u - v\|^2)$ for all $u, v$
4:     Let $N_{\text{in}}(p)$ be in-neighbors of $p$ and $N_{\text{out}}(p)$ be out-neighbors of $p$
5:     $t = \alpha \cdot (|N_{\text{in}}(p)| + |N_{\text{out}}(p)|)$
6:     $\deg(p) = \sum_{u \in N_{\text{in}}(p)} w(u, p) + \sum_{v \in N_{\text{out}}(p)} w(p, v)$
7:     **for** $u \in N_{\text{in}}(p)$ and $v \in N_{\text{out}}(p)$ **do**
8:         $w'(u, v) = w(u, v) + \frac{w(u,p) \cdot w(p,v)}{\deg(p)}$ $\triangleright$ Compute new weights for all pairs between in- and out-neighbors.
9:     **end for**
10:    $E' = \{(u, v) : u \in N_{\text{in}}(p), v \in N_{\text{out}}(p), (u, v) \in \text{Top-t}(w'(u, v))\}$
11:    $V \leftarrow V \setminus \{p\}$                              $\triangleright$ Delete $p$ from the graph.
12:    $E(p) = \{(u, v) : u \in N_{\text{in}}(p), v \in N_{\text{out}}(p)\}$
13:    $E \leftarrow E \setminus E(p) \cup E'$    $\triangleright$ Remove existing edges from $N_{\text{in}}(p)$ to $N_{\text{out}}(p)$, add edges from $E'$.
14: **end procedure**

---

**Definition 4.2.** *Let $G = (V, E, w)$. Let $W \in \mathbb{R}^{m \times m}$ be the diagonal matrix for edge weights, and let $B \in \mathbb{R}^{m \times n}$ be the signed edge-vertex incidence matrix, i.e., each row of $B$ corresponds to an edge and it has two nonzero entries on the two endpoints of the edge, with one being $+1$ and the other being $-1$ randomly assigned. The graph Laplacian matrix of $G$ is defined as $L = B^\top W B$. Equivalently, let $A \in \mathbb{R}^{n \times n}$ be its weighted adjacency matrix and $D \in \mathbb{R}^{n \times n}$ be its diagonal degree matrix, then we have $L = D - A$.*

Graph Laplacians are among the most important linear operators associated with graphs, and have a rich literature (Merris, 1994; Chung, 1997; Spielman, 2007). In the following discussion, we consider the Laplacian matrix of the clique over $N(p)$. Sparsifying by edge weights is equivalent to sparsifying by the *squared row norms* of $\sqrt{W} B$, which provides an additive error approximation in terms of the Frobenius norm (Drineas & Kannan, 2001; Frieze et al., 2004; Kannan & Vempala, 2017).

**Theorem 4.3.** *Let $G = (V, E, w)$ and $L \in \mathbb{R}^{n \times n}$ be its Laplacian matrix. Let $\varepsilon \in (0, 1)$. Suppose we generate a matrix $\widetilde{C} \in \mathbb{R}^{s \times n}$ by sampling each row of $\sqrt{W} B$ proportionally to its squared row norm with $s = 200\varepsilon^{-2}$, and reweight row $i$ by $1/(p_i s)$ where $p_i = \|(\sqrt{W} B)_{i,*}\|_2^2 / \|\sqrt{W} B\|_F^2$. Then with probability at least 0.99, $\|\widetilde{C}^\top \widetilde{C} - L\|_F \leq \varepsilon \cdot \text{tr}[W]$, where $\text{tr}[W]$ is the trace of matrix $W$.*

The proof is deferred to Appendix C. Since Theorem 4.3 sparsifies the graph by sampling edges, the resulting matrix $\widetilde{C}^\top \widetilde{C}$ also forms a weighted graph, which we denote by $G' = (V, E', w')$. Compared to the standard spectral sparsification of graphs (Spielman & Srivastava, 2011), our sparsification scheme does not necessarily preserve the *connectivity* of the resulting graph $G'$. Intuitively, sampling by edge weights is likely to ensure a cluster of close points to be well-connected, but it falls short when there are intercluster edges with small edge weights. However, we only perform the sparsification process on the *bottom layer* of HNSW, whose role is to search through edges within a cluster, after intercluster connections have been effectively handled by top layers. This is also confirmed by our experiments (see Section 5). Hence, we assume the sparsified graph $G'$ remains connected.

What does a Frobenius norm error approximation of the Laplacian imply for random walks? We prove that it approximately preserves the *hitting time* of the random walk.

**Definition 4.4.** *Let $G = (V, E, w)$ be a graph and $u, v \in V$. The* hitting time *from $u$ to $v$, denoted by $h_G(u, v)$, is the expected number of steps for a random walk starting at $u$ to reach $v$. When $G$ is clear from context, we denote it by $h(u, v)$.*

Our main result is that an additive Frobenius norm error approximation of Laplacian gives a multiplicative-additive error approximation on the hitting time.

**Theorem 4.5** (Informal version of Theorem B.8)**.** *Let $G = (V, E, w)$ be a graph, let $G' = (V, E', w')$ be the graph induced by Theorem 4.3, and suppose $G'$ is connected. Let $d_{\min}, d_{\max}$ be the min and*

*max degree of G, and let $\phi(G)$ be the edge expansion of the graph G. For any $u \neq v \in V$,*

$$|h_G(u,v) - h_{G'}(u,v)| \leq \frac{\varepsilon \operatorname{tr}[W]}{d_{\min}} h_G(u,v) + \varepsilon \operatorname{tr}[W]^2 \left( \frac{\phi(G)^2}{d_{\max}} - \varepsilon \operatorname{tr}[W] \right)^{-2}$$

*holds with probability at least 0.99, where $\phi(G) = \min_{S \subseteq V} \frac{\sum_{u \sim v, u \in S, v \notin S} w(u,v)}{\min\{|S|, |V|-|S|\}}$.*

If our softmax walk algorithm was to perform its random walk without stopping at local minima, then Theorem 4.5 would say that if $u$ is the entry point and $v$ is the desired destination, then the expected numbers of steps to reach $v$ from $u$ before and after sparsification are similar. However, the softmax walk stops at local minima, therefore, a bound on hitting time *does not* imply the walk could hit the correct destinations. Our experiments (Section 5) suggest that instead of directly correlating to the correctness of the algorithm, hitting time is a good proxy when designing the sparsification algorithm, as Algorithm 1 has good recall, query speed, deletion time, and memory utilization.

Theorem 4.3 states that to obtain an additive Frobenius norm error of $\varepsilon \cdot \operatorname{tr}[W]$, we need to sample $O(\varepsilon^{-2})$ edges, and this in turn provides an error bound on the hitting time per Theorem 4.5. How many edges do we need to sample in order to minimize the overall error? We show that in two common settings, the number of edges is linear in the number of points, $|N(p)|$:

**Corollary 4.6.** *Let $G = (V, E, w)$ be a weighted complete graph with $|V| = n$ and $G' = (V, E', w')$ be the induced graph by applying Theorem 4.3 to G. If $|E'| = O(\max_{u,v \in N(p)} h_G(u,v) \cdot n)$, then with probability at least 0.99, for any $u, v \in V$, $|h_G(u,v) - h_{G'}(u,v)| \leq \sqrt{n \cdot h_G(u,v)}$. given one of the two settings:*

- *Single cluster: for any $u, v \in V$, $w(u,v) = O(1)$;*
- *Many small clusters: there are $\sqrt{n}$ clusters of size $\sqrt{n}$. Within each cluster, the edge weights are $O(1)$, and between clusters, the edge weights are $O(1/n)$.*

**From Random Sampling to Top-$t$ Selection.** In essence, the `SPatch` framework suggests a novel approach for performing local reconnect: instead of using the distances between points in $N(p)$ directly, one should incorporate the distance between $N(p)$ and $p$ as well. This offers a natural transition to a more practical and efficient *deterministic* deletion algorithm: first compute the new local edge weights $w'(u,v)$ for all $u, v \in N(p)$, then keep the top-$t$ edges with the largest weights. Note that for large enough $r$, the probability of sampling edges outside of the top-$t$ edges is exponentially small. Thus, we could safely replace the "sample $t$ edges" step with "keep the top-$t$ edges with largest weights". This switch offers several practical advantages: in general, computing the edge weights then selecting the top-$t$ heaviest edges is more efficient than sampling $t$ edges without replacement. Thus, all our experiments are performed with the deterministic deletion algorithm.

## 5 EXPERIMENTS

We conduct extensive experiments to test the practical performance of our deletion algorithm. In the following, we will give a preliminary overview of the experimental setups, then we focus on discussing two sets of experiments: the major focus is on a *mass deletion experiment* where points are gradually deleted with no new points inserted. We wrap up the experiment by showing that the random walk search algorithm performs as well as the HNSW greedy search. Due to space constraints, we defer more details to Appendix E.

### 5.1 SETUP

**Hardware.** All experiments run on 8×3.7 GHz AMD EPYC 7R13 cores with 64 GiB RAM.

**Implementation.** We implement the HNSW data structure by utilizing the `FAISS` library (Johnson et al., 2019; Douze et al., 2024). In particular, we construct the HNSW graph by invoking the `HNSWFlatIndex` of `FAISS` with the degree parameter $m = 32$. We then extract the resulting graphs, which are directed, and convert them into a sequence of undirected `networkx` graphs (Hagberg et al., 2008) for our deletion operations.

**Datasets.** We use 4 datasets for our ANN benchmarks: `SIFT` (Jégou et al., 2011), `GIST` (Sandhawalia & Jégou, 2010), and one embedding of the `MS MARCO` (Bajaj et al., 2016) dataset with each

of `MPNet` (Song et al., 2020) and `MiniLM` (Wang et al., 2020). All of the datasets contain 1M points, with dimensions 768 (`MPNet`), 128 (`SIFT`), 960 (`GIST`) and 384 (`MiniLM`).

**Deletion algorithm.** For all deletion algorithms, we adopt the strategy that uses the tombstone for all top layers and only performs the deletion at the bottom layer. This is viable as the bottom layer is much more connected and denser than all top layers. For `SPatch`, we implement a modified version in the following aspects: 1). Since the graph is directed, we let $L$ be the set of in-neighbors of $p$ and let $R$ be the set of out-neighbors, for each $u \in R$, we pick the top-$t := \alpha \cdot \lceil \frac{|L|+|R|}{|R|} \rceil$ points $v \in L$ where $w'(u, v)$ is maximized. 2). Instead of replacing all edges in $N(p)$ including those exist before deleting $p$, we continue to add new edges with large weights until all top-$t$ edges are added. This slightly densifies the graph without adding many edges.

**Evaluation metrics.** We focus on four important evaluation metrics: top-10 recall (defined as the fraction of top-10 nearest neighbors returned by the data structure over the top-10 true nearest neighbors), number of distance computations for queries, the total time for deletion and the number of edges at the bottom layer. We use the number of distance computations as a metric for query throughput because for either the greedy search or random walk-based search, the main runtime bottleneck is the number of distance computations. For deletion procedure, as it involves more complicated operations such as computing edge weights for a clique, we directly measure the overall runtime of deletion. Finally, we use the number of edges at the bottom layer as a proxy measurement for the memory/space usage of the deletion algorithm, and in Appendix E.3 in particular Figure 7, we show that it directly corresponds to the reduction in empirical memory utilization.

## 5.2 DELETION EXPERIMENT

In this experiment, we evaluate the performance of our deletion algorithm under the mass deletion setting, as follows: In total 80% of the points will be removed from the dataset, for which every 0.8% of the points being deleted, we run the query through the remaining dataset and record the following 4 metrics: top-10 recall, number of distance computations, overall deletion time and the number of edges at the bottom layer. For the query phase, we randomly pick 5,000 query points for `SIFT`, `MPNet`, `MiniLM` and 1,000 query points for `GIST`. These queries are fixed throughout the process.

We compare our algorithm against several popular deletion prototypes for HNSW: 1). No patching, where the point is deleted from the graph without rerouting or adding any new edges. 2). Tombstone, where the point to be deleted is marked as a tombstone vertex without being deleted, and subsequent queries do not get stuck at a tombstone vertex. Xu et al. (2023) adapts a version of tombstone with periodical scan and merge to ensure the freshness of the index. 3). Local reconnect (Xu et al., 2022), where for each point in $N(p)$, an edge to its nearest neighbor in $N(p)$ is added; 4). 2-hop reconnect, where the points $u \in N(p)$ are rerouted to $N(p) \cup N(u)$ then pruned. A wide array of deletion algorithms fall into this category, including Singh et al. (2021); Xu et al. (2023); Zhao et al. (2023); Xiao et al. (2024); Xu et al. (2025). We implement and test the FreshDiskANN primitive (Singh et al., 2021). We also consider a periodic rebuild strategy: every batch, we rebuild the HNSW from scratch.

We examine Figure 2 method by method.

- While Tombstone has the best recall among all methods as per Theorem 3.1 (even better than periodic rebuild), it quickly falls short in terms of query speed ($2.5-3\times$ more distance computations than `SPatch`) and its memory usage stays constant as more points are deleted;

- Recall degrades most quickly when we do not patch, despite its fast query speed, deletion time and low memory consumption;

- Local reconnect has slightly improved recall compared to no patching, yet still worse than others;

- FreshDiskANN has better recall for datasets such as `SIFT` and `MPNet`, but the deletion time is slower. Although it might be suitable when deletions are rare, in the setting of frequent deletions it is inefficient. We also include Figure 5 for a deletion time comparison without FreshDiskANN;

- `SPatch` gives the best *overall* performance among various tradeoffs. While its recall is slightly lower than periodic rebuild and Tombstone, its query speed is much faster and memory usage decreases as more points are deleted than Tombstone. `SPatch` performs deletion much faster than FreshDiskANN and periodic rebuild.

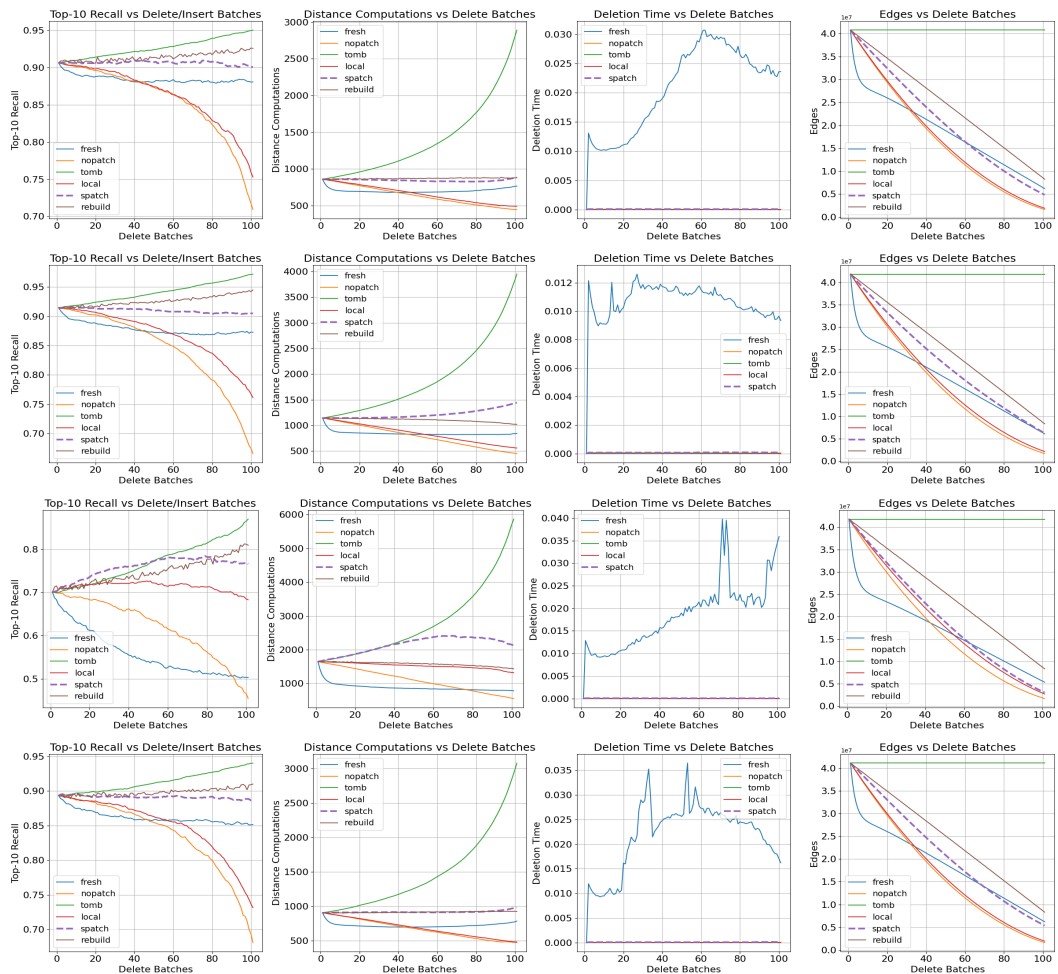

Figure 2: The rows are `MPNet`, `SIFT`, `GIST` and `MiniLM`, the columns are top-10 recall, number of distance computations per query, total deletion time and number of edges in the bottom layer of the graph. Legends: `spatch` – our algorithm `SPatch`, `fresh` – FreshDiskANN, `tomb` – tombstone, `nopatch` – no patching, local – local reconnect. For `MPNet`: we also include `rebuild` without plotting its deletion time..

## 5.3 RANDOM SOFTMAX WALK VS. GREEDY SEARCH

To empirically substantiate the validity of our randomized "twin" formulation of HNSW, we compare the two variants (softmax vs. greedy walk) in Table 2. The results show that they are highly similar empirically in recall and throughput, with the softmax variant incurring only a slight loss. This validates our use of it as a theoretical model for HNSW.

| Dataset | Top-10 Recall | | Distance Computations | |
|---------|-------|-------|-------|-------|
| | **Softmax** | **Greedy** | **Softmax** | **Greedy** |
| `MPNet` | 89.98% | 91.60% | 1562 | 1722 |
| `SIFT` | 91.13% | 91.95% | 1049 | 1109 |
| `GIST` | 71.38% | 72.46% | 1606 | 1653 |
| `MiniLM` | 89.27% | 90.09% | 1708 | 1802 |

Table 2: Comparison of softmax walk to greedy walk.

## 5.4 ADDITIONAL EXPERIMENTS

We perform additional experiments to provide a comprehensive evaluation of `Spatch` against other popular deletion methods. In Section E.5, we verify that `Spatch` our exhibits good performance on DiskANN graphs, in Figure 9, one could see that it achieves similar recall as FreshDiskANN, but has much smaller deletion time. In Section E.6, we compare `Spatch` with Dynamic Exploration Graph (DEG) (Hezel et al., 2025), yet another deletion algorithm for HNSW on `SIFT`. Through Figure 10, one could see that while DEG achieves slightly higher recall, the deletion time of DEG is much slower than ours.

## 6 CONCLUSION

In this work, we provide a theoretical framework for HNSW using random walks and sampling. We then propose a deletion algorithm, `SPatch`, based on a deterministic implementation of the theoretical motivation. Theoretical guarantees and empirical evidence demonstrate that `SPatch` has good recall, speed, deletion time and memory. This random walk interpretation opens up new opportunities to study HNSW through the lens of random walks. We hope this framework sheds light on more theoretical investigations of HNSW and inspires novel practical algorithms.

## ETHICS STATEMENT

This work designs fundamental algorithms for ANN data structures with experiments performed on standard datasets. We don't foresee any potential ethics concerns of our work.

## REPRODUCIBILITY STATEMENT

This work consists both theoretical and empirical results, we include the proofs of all theoretical results in Appendix C.

## ACKNOWLEDGMENT

We would like to thank anonymous ICLR reviewers for their helpful comments. This work is done while Lichen Zhang was interning at Amazon. Lichen Zhang is supported by a Mathworks Fellowship and a Simons Dissertation Fellowship in Mathematics.

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

APPENDIX

**Roadmap.** In Section A, we provide more preliminaries regarding notations and inequalities used throughout the paper. In Section B, we show how to obtain a multiplicative-additive approximation for hitting time given an additive Frobenius norm approximation of the Laplacian. In Section C, we provide missing proofs for prior statements. In Section D, we supply more details for HNSW data structure. In Section E, we present more details about experiments in Section 5 and an extra set of experiments in the steady state setting, experiments on DiskANN and extra comparison with other deletion strategies.

## A MORE PRELIMINARIES

### A.1 NOTATIONS

Let $n$ be a natural number, we use $[n]$ to denote the set $\{1, 2, \ldots, n\}$. Let $X$ be a random variable, we use $\mathbb{E}[X]$ to denote the expectation of $X$, and let $E$ be an event, we use $\Pr[E]$ to denote the probability that event $E$ happens.

Let $G = (V, E, w)$ be a graph where $V$ is the set of vertices, $E$ is the set of edges and $w : E \to \mathbb{R}_+$ be the weight function that assigns a positive real number to each edge. We adopt the convention and let $n := |V|$ and $m := |E|$. Given a vertex $u \in V$, we use $N(u)$ to denote its neighborhood, i.e., $N(u) = \{v \in V : \{u, v\} \in E\}$.

Let $x \in \mathbb{R}^d$, without specification, we use $\|x\|$ to denote a general norm of $x$, we use $\|x\|_2$ to denote the $\ell_2$ or Euclidean norm of $x$. Let $M \in \mathbb{R}^{n \times d}$ be a matrix, we use $\|M\|$ to denote the spectral norm of $M$, $\|M\|_F$ to denote its Frobenius norm. We use $M^\dagger$ to denote the pseudo-inverse of matrix $M$. If $M \in \mathbb{R}^{n \times n}$ and is real, symmetric, we use $\lambda_1(M), \ldots, \lambda_n(M)$ to denote its eigenvalues, ordered in ascending order. When $M$ is clear from context, we use $\lambda_1, \ldots, \lambda_n$ to denote these eigenvalues. We use $\mathrm{tr}[M]$ to denote the trace of $M$, i.e., $\mathrm{tr}[M] = \sum_{i=1}^n M_{i,i}$.

### A.2 USEFUL INEQUALITIES

We collect some useful inequalities to be used later.

**Lemma A.1** (Markov's inequality). *Let $X$ be a non-negative random variable and $a > 0$, then*

$$\Pr[X > a \cdot \mathbb{E}[X]] \leq \frac{1}{a}.$$

**Lemma A.2** (Weyl's inequality, Weyl (1912)). *Let $A, B \in \mathbb{R}^{n \times n}$ be symmetric matrices, then for any $i \in [n]$,*

$$|\lambda_i(A) - \lambda_i(B)| \leq \|A - B\|.$$

**Lemma A.3** (Theorem 4.1 of Wedin (1973)). *For two conforming matrices $A, B$,*

$$\|A^\dagger - B^\dagger\| \leq 2 \cdot \max\{\|A^\dagger\|^2, \|B^\dagger\|^2\} \cdot \|A - B\|.$$

## B HITTING TIME AND MULTIPLICATIVE-ADDITIVE APPROXIMATION

In this section, we prove that if we can generate a matrix $L'$ with $\|L' - L\|_F \leq \delta$, then the hitting time can be also approximated in a multiplicative-additive error manner. We first introduce effective resistance, an important metric on graphs that could be associated with the graph Laplacian matrix.

**Definition B.1.** *Let $G = (V, E, w)$ be a graph and $u, v \in V$, the* effective resistance *between $u$ and $v$, denoted by $R_G(u, v)$, is defined as*

$$R_G(u, v) = \chi_{u,v}^\top L^\dagger \chi_{u,v},$$

*where $\chi_{u,v} = e_u - e_v$.*

Tetali (Tetali, 1991) proves that hitting time and effective resistance are intrinsically connected as follows.

**Lemma B.2** (Theorem 5 of Tetali (1991)). *Let $G = (V, E, w)$ be a graph and $u, v \in V$ with $u \neq v$, the hitting time and effective resistance obeys the following identity:*

$$h(u, v) = \frac{1}{2} \sum_{z \in V} \deg(z)(R_G(u, v) + R_G(v, z) - R_G(u, z)).$$

We need to define the edge expansion of a graph, as we will use Cheeger's inequality to lower bound the smallest nontrivial eigenvalue of the Laplacian matrix.

**Definition B.3** (Edge expansion). *Let $G = (V, E, w)$ be a graph and $S \subseteq V$, define $e(S) = \sum_{u \sim v, u \in S, v \in V \setminus S} w(u, v)$, the* edge expansion *of a set $S$ is defined as*

$$\phi(S) = \frac{e(S)}{\min\{|S|, |V \setminus S|\}},$$

*the* edge expansion *of the graph $G$ is defined as*

$$\phi(G) = \min_{S \subseteq V} \phi(S).$$

Cheeger's inequality gives a lower bound on $\lambda_2(L)$ in terms of edge expansion.

**Lemma B.4** (Cheeger's inequality, Cheeger (1971)). *Let $G = (V, E, w)$ be a graph and $\phi(G)$ be the edge expansion of $G$ (Definition B.3). Let $\lambda_2$ be the second smallest eigenvalue of $L$ and $d_{\max} = \max_{u \in V} \deg(u)$, then*

$$\frac{\phi(G)^2}{2d_{\max}} \leq \lambda_2 \leq 2\phi(G).$$

We next prove that if $\|L - L'\|_F \leq \delta$, then $\|L^\dagger - (L')^\dagger\|$ can also be bounded by invoking Lemma A.3.

**Lemma B.5.** *Let $G = (V, E, w)$ be a graph and $L$ be its corresponding Laplacian matrix, let $G' = (V, E', w')$ be the graph where $E' \subseteq E$, $L'$ be its Laplacian matrix and $G'$ is connected. Suppose $\|L - L'\|_F \leq \delta$ for some $\delta > 0$, then*

$$\|L^\dagger - (L')^\dagger\| \leq 2\delta \left(\frac{\phi(G)^2}{2d_{\max}} - \delta\right)^{-2},$$

*where $\phi(G)$ is the edge expansion of $G$ (Definition B.3) and $d_{\max}$ is the max degree of $G$.*

*Proof.* The proof will be combining Lemma A.3 and Lemma B.4. By Lemma A.3, we have

$$\|L^\dagger - (L')^\dagger\| \leq 2 \cdot \max\{\|L^\dagger\|^2, \|(L')^\dagger\|^2\} \cdot \|L - L'\|,$$

where we already have

$$\|L - L'\| \leq \|L - L'\|_F \leq \delta,$$

meanwhile, by Weyl's inequality (Lemma A.2), this also implies a bound on $\lambda_2(L')$:

$$|\lambda_2(L) - \lambda_2(L')| \leq \|L - L'\| \leq \delta,$$

it remains to establish a bound on $\lambda_2(L)$. By Cheeger's inequality, we obtain

$$\lambda_2(L) \geq \frac{\phi(G)^2}{2d_{\max}},$$

since $L$ has rank $n - 1$, we know that

$$\|L^\dagger\| = \frac{1}{\lambda_2(L)}$$

$$\leq \frac{2d_{\max}}{\phi(G)^2},$$

similarly we can attempt to establish a bound on $\lambda_2(L')$:

$$\lambda_2(L) - \delta \leq \lambda_2(L') \leq \lambda_2(L) + \delta,$$

therefore

$$\lambda_2(L') \geq \frac{\phi(G)^2}{2d_{\max}} - \delta$$

and by the same argument as $\|L^\dagger\|$,

$$\|(L')^\dagger\| = \frac{1}{\lambda_2(L')}$$

$$\leq \frac{2d_{\max}}{\phi(G)^2 - 2d_{\max}\delta},$$

we conclude the following bound:

$$\max\{\|L^\dagger\|^2, \|(L')^\dagger\|^2\} \leq \left(\frac{\phi(G)^2}{2d_{\max}} - \delta\right)^{-2}.$$

Put things together, we obtain the final bound:

$$\|L^\dagger - (L')^\dagger\| \leq 2\delta \left(\frac{\phi(G)^2}{2d_{\max}} - \delta\right)^{-2}. \qquad \square$$

As a natural corollary, we also obtain a bound on the effective resistance.

**Corollary B.6.** *Let $G = (V, E, w)$ be a graph and $G' = (V, E', w')$ with $E' \subseteq E$, $L'$ be its Laplacian matrix and $G'$ is connected. Suppose $\|L - L'\|_F \leq \delta$ for some $\delta > 0$, then for any $u, v \in V$,*

$$|R_G(u, v) - R_{G'}(u, v)| \leq 4\delta \left(\frac{\phi(G)^2}{2d_{\max}} - \delta\right)^{-2}.$$

*Proof.* Fix $u, v \in V$, then

$$|\chi_{u,v}^\top (L^\dagger - (L')^\dagger)\chi_{u,v}| \leq \|L^\dagger - (L')^\dagger\| \cdot \|\chi_{u,v}\|_2^2$$

$$\leq 4\delta \left(\frac{\phi(G)^2}{2d_{\max}} - \delta\right)^{-2},$$

where the last step is by invoking the bound on $\|L^\dagger - (L')^\dagger\|$ of Lemma B.5. $\qquad \square$

It would also be useful to have a handle on the degree.

**Corollary B.7.** *Let $G = (V, E, w)$ be a graph and $G' = (V, E', w')$ with $E' \subseteq E$, $L'$ be its Laplacian matrix and $G'$ is connected. Suppose $\|L - L'\|_F \leq \delta$ for some $\delta > 0$, then for any $u \in V$,*

$$|\deg_G(u) - \deg_{G'}(u)| \leq \delta.$$

*Proof.* Note that $\deg_G(u) = e_u^\top L e_u$ and $\deg_{G'}(u) = e_u^\top L' e_u$, thus

$$|\deg_G(u) - \deg_{G'}(u)| = |e_u^\top (L - L')e_u|$$

$$\leq \|L - L'\| \cdot \|e_u\|_2^2$$

$$= \|L - L'\|$$

$$\leq \delta. \qquad \square$$

We are in the position to prove our main theorem regarding hitting time.

**Theorem B.8.** *Let $G = (V, E, w)$ be a graph and $G' = (V, E', w')$ with $E' \subseteq E$, $L'$ be its Laplacian matrix and $G'$ is connected. Suppose $\|L - L'\|_F \leq \delta$ for some $\delta > 0$, then for any $u, v \in V$ with $u \neq v$,*

$$|h_G(u, v) - h_{G'}(u, v)| \leq \frac{\delta}{d_{\min}} h_G(u, v) + 12\delta \left(\frac{\phi(G)^2}{2d_{\max}} - \delta\right)^{-2} \sum_{e \in E} w_e$$

*Proof.* By Lemma B.2, we know that

$$h_G(u,v) = \frac{1}{2} \sum_{z \in V} \deg_G(z)(R_G(u,v) + R_G(v,z) - R_G(u,z)), \tag{1}$$

by Corollary B.6, we have that

$$|R_G(u,v) - R_{G'}(u,v)| \le 4\delta \left( \frac{\phi(G)^2}{2d_{\max}} - \delta \right)^{-2},$$

for ease of notation, let $\delta' := 4\delta \left( \frac{\phi(G)^2}{2d_{\max}} - \delta \right)^{-2}$, and by Corollary B.7,

$$|\deg_G(u) - \deg_{G'}(u)| \le \delta.$$

To apply Eq. (1), we examine one term as follows:

$$\begin{aligned}
\deg_G(z)R_G(u,v) - \deg_{G'}(z)R_{G'}(u,v) &\le \deg_G(z)R_G(u,v) - (\deg_G(z) - \delta)R_{G'}(u,v) \\
&\le \deg_G(z)R_G(u,v) - (\deg_G(z) - \delta)(R_G(u,v) - \delta') \\
&= \delta' \deg_G(z) + \delta R_G(u,v) - \delta\delta' \\
&\le \delta' \deg_G(z) + \delta R_G(u,v)
\end{aligned}$$

Putting it together yields

$$\begin{aligned}
&h_G(u,v) - h_{G'}(u,v) \\
&= \frac{1}{2} \sum_{z \in V} (\deg_G(z)(R_G(u,v) + R_G(v,z) - R_G(u,z)) - \deg_{G'}(z)(R_{G'}(u,v) + R_{G'}(v,z) - R_{G'}(u,z))) \\
&\le \frac{1}{2} \sum_{z \in V} 3\delta' \deg_G(z) + \delta(R_G(u,v) + R_G(v,z) - R_G(u,z)) \\
&= \frac{3}{2}\delta' \sum_{z \in V} \deg_G(z) + \frac{1}{2}\delta \sum_{z \in V} R_G(u,v) + R_G(v,z) - R_G(u,z) \\
&\le \frac{3}{2}\delta' \sum_{z \in V} \deg_G(z) + \frac{1}{2}\delta \sum_{z \in V} \deg_G(z)(R_G(u,v) + R_G(v,z) - R_G(u,z)) \cdot \frac{1}{\deg_G(z)} \\
&\le \frac{3}{2}\delta' \sum_{z \in V} \deg_G(z) + \frac{1}{2}\delta \left( \sum_{z \in V} \deg_G(z)(R_G(u,v) + R_G(v,z) - R_G(u,z)) \right) \cdot \frac{1}{d_{\min}} \\
&= 3\delta' \sum_{e \in E} w_e + \frac{\delta}{d_{\min}} h_G(u,v),
\end{aligned}$$

this completes the proof. □

## C   MISSING PROOFS

In this section, we include the missing proofs in previous sections.

**Theorem C.1** (Restatement of Theorem 3.1). *Let $P \subset \mathbb{R}^d$ be an $n$-point dataset and $p \in P$ be a point to-be-deleted. Suppose $P$ is preprocessed by an HNSW data structure and $p$ is not the entry point of the HNSW. Fix a query point $q \in \mathbb{R}^d$ and suppose the search reaches layer $l \in \{1, \dots, L\}$, let $N(p)$ denote the neighborhood of $p$ at layer $l$. Suppose $q$ reaches $N(p)$, visits and leaves $p$. Consider the deletion procedure that removes $p$ at layer $l$ and forms a clique over $N(p)$, then the search of $q$ on the new graph is equivalent to the search of $q$ on the old graph.*

*Proof.* Let $G$ denote the graph at layer $l$ before deleting $p$ and $G_{\setminus p}$ denote the graph at layer $l$ after deleting $p$. By assumption, there is some vertex $a \in N(p)$ visited by the walk immediately before visiting $p$, and another vertex $b \in N(p)$ visited immediately after. We focus our attention on how the graph transformation affects the $a \to p \to b$ section of the traversal. In the original graph, the walk transitioned from $a$ to $p$ because $p$ is the vector in $N(a)$ nearest to $q$. However, since the walk then

transitions from $p$ to $b$, it must be the case that $b$ is closer to $q$ than is any point in $N(p) \cup \{p\}$, and thus $\|b - q\| = \min_{c \in N(p)} \|c - q\| \leq \|p - q\|$.

Now, consider the new graph where $p$ is deleted and a clique is instead inserted between the vectors in its former neighborhood. The walk remains the same until it first hits $a$. From $a$, there are two possible (not mutually-exclusive) types of neighbors the walk could transition to: those that were neighbors of $a$ in the original graph, and those new neighbors it acquired when the clique was inserted on $N(p)$, including $b$. Because the old walk transitioned into $p$, it must be the case that $\|p - q\| \leq \|x - q\|$ for each $x$ among the original $N(a)$. Combining this with the previous inequality, it must be the case that $\|b - q\| = \min_y \|y - q\|$, with $y$ ranging over the entire *new* neighborhood of $a$, and thus the walk must still transition into $b$, at which point it proceeds as before. $\qquad \square$

**Theorem C.2** (Restatement of Theorem 4.1). *Let $G = (V, E, w)$ be a weighted graph, define the random walk under the weights $w$ for any edge $\{u, v\} \in E$ as $\Pr[u \to v \mid w] = \frac{w(u,v)}{\deg(u)}$ where $\deg(u) = \sum_{z \in N(u)} w(u, z)$. Let $p$ be the point to be deleted as for any $u, v \in N(p)$, define the new weights $w'(u, v)$ as $w'(u, v) = w(u, v) + \frac{w(u,p) \cdot w(p,v)}{\deg(p)}$, let $E(p) = \{\{u, p\} : u \in N(p)\}$ and $C(p) = \{\{u, v\} : u \neq v, u, v \in N(p)\}$, then for the new graph $G' = (V \setminus \{p\}, E \setminus E(p) \cup C(p), w')$, we have*

$$\Pr[u \to v \mid w'] = \Pr[(u \to p \to v) \vee (u \to v) \mid w].$$

*Proof.* Consider the neighborhood of $p$, $N(p)$, let $u, v \in N(p)$, we reason over the probability that the walk moves from $u$ to $v$. Note that after deletion, the only change is that the vertex $p$ has been removed from the graph, therefore, there is no path from $u \to p \to v$. On the other hand, there is now a direct path from $u$ to $v$ under the new weight $w'(u, v)$, so we need to show that the probability is not affected.

Recall that for any vertex $z \in N(u)$, we have that the probability the walk moves from $u$ to $z$ is

$$\frac{w(u, z)}{\deg(u)}$$

where $\deg(u) = \sum_{z \in N(u)} w(u, z)$. Before deleting $p$, we calculate the probability of the path $u \to p \to v$ together with the probability of $u \to v$:

$$
\begin{aligned}
&\Pr[(u \to p \to v) \vee (u \to v) \mid w] \\
&= \Pr[u \to p \mid w] \cdot \Pr[p \to v \mid u \to p, w] + \Pr[u \to v \mid w] \\
&= \frac{w(u, p)}{\deg(u)} \cdot \frac{w(p, v)}{\deg(p)} + \frac{w(u, v)}{\deg(u)}
\end{aligned}
$$

After deletion, the random walk is performed via new weights $w'$:

$$
\begin{aligned}
\Pr[u \to v \mid w'] &= \frac{w'(u, v)}{\deg'(u)} \\
&= \frac{w(u, v)}{\deg'(u)} + \frac{w(u, p) \cdot w(p, v)}{\deg(p) \deg'(u)}
\end{aligned}
\tag{2}
$$

note that

$$
\begin{aligned}
\deg'(u) &= \sum_{v \in N(u) \setminus N(p)} w(u, v) + \sum_{v \in N(p)} w'(u, v) \\
&= \deg(u) - w(u, p) + \sum_{v \in N(p)} \frac{w(u, p) \cdot w(p, v)}{\deg(p)} \\
&= \deg(u) - w(u, p) + w(u, p) \\
&= \deg(u)
\end{aligned}
$$

therefore

$$(2) = \frac{w(u, v)}{\deg(u)} + \frac{w(u, p) \cdot w(p, v)}{\deg(p) \cdot \deg(u)},$$

which is the same as the probability of walking from $u$ to $v$ either via the edge $\{u, v\}$ or the path $u \to p \to v$. $\qquad \square$

**Theorem C.3** (Restatement of Theorem 4.3). *Let $G = (V, E, w)$ and $L \in \mathbb{R}^{n \times n}$ be its graph Laplacian matrix. Suppose we generate a matrix $\widetilde{C} \in \mathbb{R}^{s \times n}$ by sampling each row of $\sqrt{W}B$ proportional to its squared row norm with $s = 200\varepsilon^{-2}$, and reweight row $i$ by $1/(p_i s)$ where $p_i = \|(\sqrt{W}B)_{i,*}\|_2^2 / \|\sqrt{W}B\|_F^2$, then with probability at least 0.99,*

$$\|\widetilde{C}^\top \widetilde{C} - L\|_F \leq \varepsilon \cdot \text{tr}[W].$$

*Proof.* For simplicity of notation, we let $C := \sqrt{W}B$. Define the random variable $X_i = \frac{1}{p_i} C_{i,*} C_{i,*}^\top$, where $p_1, \ldots, p_m$ are the sampling probabilities of the process, i.e. $p_i = \|C_{i,*}\|_2^2 / \|C\|_F^2$. We prove several important properties of these $X_i$'s.

- Expectation. Note that

$$
\begin{aligned}
\mathbb{E}[X] &= \sum_{i=1}^m p_i \cdot \frac{1}{p_i} C_{i,*} C_{i,*}^\top \\
&= \sum_{i=1}^m C_{i,*} C_{i,*}^\top \\
&= C^\top C
\end{aligned}
$$

- Expected Frobenius norm. We compute the entrywise variance of $X$:

$$
\begin{aligned}
&\mathbb{E}[\|X\|_F^2] \\
&= \sum_{i,j=1}^n \mathbb{E}[x_{i,j}^2] \\
&= \left( \sum_{i,j=1}^n \sum_{k=1}^n p_k \frac{1}{p_k^2} \cdot C_{k,i}^2 C_{k,j}^2 \right) \\
&= \sum_{k=1}^n \frac{1}{p_k} \|C_{k,*}\|_2^4 \\
&= \|C\|_F^4,
\end{aligned}
$$

let $Y = \frac{1}{s} \sum_{i=1}^s X_i$, then

$$
\begin{aligned}
&\mathbb{E}[\|Y\|_F^2] \\
&= \mathbb{E}[\|\frac{1}{s} \sum_{i=1}^s X_i\|_F^2] \\
&= \frac{1}{s^2} \left( \sum_{i=1}^s \mathbb{E}[\|X\|_F^2] + 2 \sum_{i \neq j} \mathbb{E}[\text{tr}[X_i X_j]] \right) \\
&= \frac{\|C\|_F^4}{s} + \frac{2}{s^2} \sum_{i \neq j} \text{tr}[\mathbb{E}[X_i X_j]] \\
&= \frac{\|C\|_F^4}{s} + \frac{2}{s^2} \sum_{i \neq j} \text{tr}[\mathbb{E}[X_i] \mathbb{E}[X_j]] \\
&= \frac{\|C\|_F^4}{s} + \frac{s-1}{s} \|C^\top C\|_F^2.
\end{aligned}
$$

- Probability. We will be using Markov inequality on $\|Y - C^\top C\|_F^2$, to do so we first compute

$$
\begin{aligned}
\mathbb{E}[\text{tr}[YC^\top C]] &= \text{tr}[\mathbb{E}[YC^\top C]] \\
&= \text{tr}[\mathbb{E}[Y]C^\top C]
\end{aligned}
$$

$$= \mathrm{tr}[C^\top C C^\top C]$$
$$= \|C^\top C\|_F^2,$$

and we can compute the expectation of the squared Frobenius norm deviation:

$$\mathbb{E}[\|Y - C^\top C\|_F^2]$$
$$= \mathbb{E}[\|Y\|_F^2] + \|C^\top C\|_F^2 - 2\,\mathbb{E}[\mathrm{tr}[Y C^\top C]]$$
$$= \frac{\|C\|_F^4}{s} + \frac{2s-1}{s}\|C^\top C\|_F^2 - 2\|C^\top C\|_F^2$$
$$\leq \frac{\|C\|_F^4}{s}.$$

Set $s = 100\varepsilon^{-2}$, we obtain that $\mathbb{E}[\|Y - C^\top C\|_F^2] \leq \varepsilon^2\|C\|_F^4$. By Markov's inequality (Lemma A.1), we have

$$\Pr[\|Y - C^\top C\|_F^2 > \varepsilon^2\|C\|_F^4] \leq \frac{\varepsilon^2/100 \cdot \|C\|_F^4}{\varepsilon^2\|C\|_F^4} = \frac{1}{100},$$

as desired. Utilizing the structure of $C$, we could further simplify the bound: $\|C\|_F^2 = 2\sum_{e\in E} w_e = 2\|w\|_1 = 2\,\mathrm{tr}[W]$. $\qquad\square$

**Corollary C.4** (Restatement of Corollary 4.6). *Let $G = (V, E, w)$ be a weighted complete graph and $G' = (V, E', w')$ be the induced graph by applying Theorem 4.3 to $G$, and assume $G'$ is connected. If $|E'| = O(\max_{u,v\in N(p)} h_G(u,v) \cdot n)$, then with probability at least 0.99, for any $u, v \in V$, $|h_G(u,v) - h_{G'}(u,v)| \leq \sqrt{n \cdot h_G(u,v)}$. given one of the two settings:*

- *Single cluster: for any $u, v \in V$, $w(u,v) = O(1)$;*

- *Many small clusters: there are $\sqrt{n}$ clusters of size $\sqrt{n}$. Within each cluster, the edge weights are $O(1)$, and between clusters, the edge weights are $O(1/n)$.*

*Proof.* We prove the two settings item by item.

- Single cluster, that is, for all $u, v, u', v' \in V$, we have $w(u,v) = O(w(u',v')) = O(1)$. In this case, $\mathrm{tr}[W] = O(n^2)$, $d_{\min}, d_{\max} = O(n)$ and $\phi(G) = O(n)$. The multiplicative error factor for $h_G(u,v)$ is then $\varepsilon \cdot n$ and the additive error term is $\varepsilon \cdot n^4(n - \varepsilon \cdot n^2)^{-2} = \frac{\varepsilon n^2}{(1-\varepsilon n)^2} \leq 4\varepsilon^{-1}$, so the overall error is $\varepsilon n \cdot h_G(u,v) + 4\varepsilon^{-1}$, equating these two terms sets $\varepsilon^{-1} = \sqrt{n \cdot h_G(u,v)}$. According to Theorem 4.3, this means that we can sparsify the number of edges in clique $C(p)$ from $O(|N(p)|^2)$ down to $O(\max_{u,v\in N(p)} h(u,v) \cdot |N(p)|)$.

- Many small clusters, in particular the max edge weight is $O(1)$ while the min edge weight is $O(n^{-1})$. Among the $n$ points, we assume they are clustered $n^{0.5}$ parts, each of size $n^{0.5}$. For the $O(n^2)$ intercluster edges, the edge weights are $O(n^{-1})$, while other edges have weights $O(1)$. In this case, $\mathrm{tr}[W] = n + n^{1.5} \leq O(n^{1.5})$, $d_{\min}, d_{\max} = O(n^{0.5})$ and $\phi(G) = O(1)$, and the multiplicative factor is $\varepsilon \cdot n$ and the additive factor is $\varepsilon \cdot n^3(n^{-0.5} - \varepsilon \cdot n^{1.5})^{-2} = \frac{\varepsilon n^4}{(1-\varepsilon n^2)^2} \leq 4\varepsilon^{-1}$, so the overall error is $\varepsilon n \cdot h_G(u,v) + 4\varepsilon^{-1}$. We would set $\varepsilon^{-1} = \sqrt{n \cdot h_G(u,v)}$ to minimize the error. Note that now the number of edges in the sparsified graph is $O(\max_{u,v\in N(p)} h(u,v) \cdot |N(p)|)$. $\qquad\square$

# D HNSW DATA STRUCTURE

We review the both classical HNSW data structure proposed in Malkov & Yashunin (2020), also provide more details about our random walk-based variant of it in this section.

## D.1 CLASSICAL HNSW

In this section, we provide a more in-depth review of the HNSW data structure. We lay out its structure in several algorithms, starting from the search procedure.

---

**Algorithm 2** HNSW algorithm: layer search.

---

1: **procedure** LAYERSEARCH($q \in \mathbb{R}^d, P \subset \mathbb{R}^d, l \in \{1, \ldots, L\}, u \in P, m \in [|P|]$)
2:          ▷ $l$ is the layer of the graph, $u$ is the starting point and $m$ is the total number of nearest neighbors to return.
3:   `visited` $\leftarrow \{u\}$                                                   ▷ Visited vertices.
4:   `candidates` $\leftarrow \{u\}$                                          ▷ Candidate vertices.
5:   `nbrs` $\leftarrow \{u\}$                                    ▷ Dynamic list of nearest neighbors.
6:   **while** $|$`candidates`$| > 0$ **do**
7:      $c \leftarrow$ nearest neighbor of $q$ in `candidates`
8:      $f \leftarrow$ furthest neighbor of $q$ in `nbrs`
9:      `candidates` $\leftarrow$ `candidates` $\setminus \{c\}$
10:      **if** $\|c - q\| > \|f - q\|$ **then**
11:         **break**
12:      **end if**
13:      **for** $v \in$ nbrhood$(c)$ at layer $l$ **do**
14:         **if** $v \notin$ `visited` **then**
15:            `visited` $\leftarrow$ `visited` $\cup \{v\}$
16:            $f \leftarrow$ furthest neighbor of $q$ in `nbrs`
17:            **if** $\|v - q\| < \|f - q\|$ or $|$`nbrs`$| < m$ **then**   ▷ Either $v$ is closer or `nbrs` is not full.
18:               `candidates` $\leftarrow$ `candidates` $\cup \{v\}$
19:               `nbrs` $\leftarrow$ `nbrs` $\cup \{v\}$
20:               **if** $|$`nbrs`$| > m$ **then**
21:                  Remove furthest neighbor of $q$ in `nbrs`
22:               **end if**
23:            **end if**
24:         **end if**
25:      **end for**
26:   **end while**
27:   **return** `nbrs`
28: **end procedure**

---

To construct the HNSW data structure, we implement the insertion procedure.

The insertion works as follows: it simply performs Algorithm 2 from $L$ to $l + 1$ where $l$ is the designated layer for $q$ to be inserted. Starting from layer $l$ to 1, we increase the number of points to be returned by Algorithm 2 (efConstruction can be much larger $m$) and then select $m$ of them to add edges using NEIGHBORSELECT. This procedure is then repeated for the new neighbors of $q$ to prune edges.

The only missing piece is the neighbor selection procedure. Malkov & Yashunin (2020) recommends two types of neighbor selection: one is simply taking the top-$m$ nearest neighbors, while the other involves a more sophisticated heuristic procedure. We refer readers to Malkov & Yashunin (2020) for more details.

### D.2   PROBABILISTIC HNSW

Inspired by the classical HNSW data structure, we propose a random walk-based approach for both searching and constructing the data structure. While we give an overview in Section 4, here we provide the complete algorithm.

The key distinction between Algorithm 2 and 4 is on line 7: instead of taking the nearest neighbor, Algorithm 4 samples a point to move to based on the softmax of negative squared distance. For insertion, we give an alternative presentation that shows how to construct the one layer of HNSW by first building a complete graph, then randomly sparsifying it. It is showcases in Algorithm 5 and Figure 3.

---

**Algorithm 3** HNSW algorithm: insertion.

---

1: **procedure** INSERT($q \in \mathbb{R}^d, P \subset \mathbb{R}^d, u \in P, \texttt{efConstruction} \in [|P|], m \in [|P|], m_{\max} \in [|P|]$)
2:     $l \leftarrow \lfloor -\ln(\text{Unif}(0,1))/\ln m \rfloor$
3:     **for** $l_c = L \rightarrow l + 1$ **do**
4:         $\texttt{candidates} \leftarrow$ LAYERSEARCH$(q, P, l_c, u, 1)$
5:         $u \leftarrow \arg\min_{v \in \texttt{candidates}} \|v - q\|$
6:     **end for**
7:     **for** $l_c = \min\{L, l\} \rightarrow 1$ **do**
8:         $\texttt{candidates} \leftarrow$ LAYERSEARCH$(q, P, l_c, u, \texttt{efConstruction})$
9:         $\texttt{nbrs} \leftarrow$ NEIGHBORSELECT$(q, \texttt{candidates}, m, l_c)$
10:       Add edges between $q$ and $\texttt{nbrs}$ at layer $l_c$
11:     **end for**
12:     **for** $v \in \texttt{nbrs}$ **do**
13:         $N(v) \leftarrow \text{nbrhood}(v)$ at layer $l_c$
14:         **if** $|N(v)| > m_{\max}$ **then**
15:            $\texttt{newNbrs} \leftarrow$ NEIGHBORSELECT$(v, N(v), m_{\max}, l_c)$
16:            Add edges between $v$ and $\texttt{newNbrs}$ at layer $l_c$
17:         **end if**
18:     **end for**
19:     $u \leftarrow \texttt{candidates}$
20:     **if** $l > L$ **then**
21:         $u \leftarrow q$
22:     **end if**
23: **end procedure**

---

## D.3 RUNTIME ANALYSIS OF SPATCH

We provide a preliminary runtime analysis of SPatch (Algorithm 1). Let $p$ be the point we want to delete, then note that the most expensive operation is to compute and update $w'(u, v)$, which takes $O(|N_{\text{in}}(p)| \cdot |N_{\text{out}}(p)| \cdot d)$ time, all other operations are subsumed by this step.

## D.4 CONSTRUCTION VIA SPARSIFICATION

The typical construction process of an HNSW graph involves the following steps:

- Given a point $u$, determine the layer $l_u$ to be inserted;
- Perform greedy search for $u$ from layer $L$ down to $l_u + 1$, moving a layer downward each time the search is stuck;
- In layers $l = l_u \ldots 1$, perform a greedy search for $u$, and draw edges between $u$ and the $m$ nearest points to it that were visited along the search path.

Having replaced greedy search with a softmax walk, we can provide a probabilistic interpretation of the graph construction process as a sparsification of the weighted complete graph, as follows. Let $G_0$ be a weighted complete graph with the edge weight between two vertices $u, v$ being $\exp(-r^2 \cdot \|u - v\|^2)$. We claim that each layer of the HNSW graph $G_1$ can be constructed via a random walk-based sparsification of $G_0$. To this end, fix an ordering of the vertices $v_1, \ldots, v_n$, and initialize the graph with a single node $v_1$. Then, alternate between the complete graph $G_0$ and a sparsified graph as follows: for $i = 2, \ldots, n$,

- **Densify**: Add $v_i$ to $G_1$ by adding all edge $\{v_i, v_j\}$ for $j \in [i - 1]$, with edge weights $\exp(-r^2 \cdot \|v_i - v_j\|^2)$.
- **Random walk**: Perform a softmax walk for $v_i$ in $G_1$ starting at $v_1$, maintaining a list of the nodes $\texttt{visited}$ along the walk.
- **Sparsify**:
  - After the random walk terminates, sample $m$ points from $\texttt{visited}$ with probability proportional to the edge weights attached to them in the Densify step, forming a set $\texttt{sample}$. Remove edges between $v_i$ and points not in $\texttt{sample}$.

---

**Algorithm 4** HNSW algorithm: layer search.

---

1: **procedure** LAYERSEARCHRANDOMWALK($q \in \mathbb{R}^d, P \subset \mathbb{R}^d, l \in \{0, \ldots, L\}, u \in P, m \in [|P|], r \in \mathbb{R}_+$)
2:                $\triangleright$ $l$ is the layer of the graph, $u$ is the starting point and $m$ is the total number of nearest neighbors to return.
3:     visited $\leftarrow \{u\}$                              $\triangleright$ Visited vertices.
4:     candidates $\leftarrow \{u\}$                        $\triangleright$ Candidate vertices.
5:     nbrs $\leftarrow \{u\}$                          $\triangleright$ Dynamic list of nearest neighbors.
6:     **while** $|$candidates$| > 0$ **do**
7:         Sample $c$ with probability $\frac{\exp(-r^2 \cdot \|c-q\|^2)}{\sum_{v \in \text{candidates}} \exp(-r^2 \cdot \|v-q\|^2)}$
8:         $f \leftarrow$ furthest neighbor of $q$ in nbrs
9:         candidates $\leftarrow$ candidates $\setminus \{c\}$
10:        **if** $\|c - q\| > \|f - q\|$ **then**
11:            **break**
12:        **end if**
13:        **for** $v \in$ nbrhood$(c)$ at layer $l$ **do**
14:            **if** $v \notin$ visited **then**
15:                visited $\leftarrow$ visited $\cup \{v\}$
16:                $f \leftarrow$ furthest neighbor of $q$ in nbrs
17:                **if** $\|v - q\| < \|f - q\|$ or $|$nbrs$| < m$ **then**   $\triangleright$ Either $v$ is closer or nbrs is not full.
18:                    candidates $\leftarrow$ candidates $\cup \{v\}$
19:                    nbrs $\leftarrow$ nbrs $\cup \{v\}$
20:                    **if** $|$nbrs$| > m$ **then**
21:                        Remove furthest neighbor of $q$ in nbrs
22:                    **end if**
23:                **end if**
24:            **end if**
25:        **end for**
26:     **end while**
27:     **return** nbrs
28: **end procedure**

---

- Sparsify the neighborhood of each $u \in$ sample by subsampling $m$ of the edges incident to $u$ with probability to their edge weight.

This point of view also justifies our use of edge weights when performing deletion, as we could treat the edge weights are formed during construction and inserting the corresponding the point.

# E   ADDITIONAL EXPERIMENTS

## E.1   DETAILS OF PREVIOUS EXPERIMENTS

We start by examining more details regarding prior experiments conducted in Section 5.

**Deletion experiments.** Regarding deletion algorithms, we note that several of them have hyper-parameters:

- FreshDiskANN (Singh et al., 2021) requires a hyper-parameter $\alpha$, which governs how many edges to prune after rerouting $u$ to $N(p) \cup N(u)$, intuitively, the larger the $\alpha$, the denser the graph. According to Singh et al. (2021), $\alpha$ should be chosen $> 1$. In our experiment, due to the time-consuming nature of FreshDiskANN deletion procedure, we set $\alpha = 1.2$. It is also worth noting that FreshDiskANN is designed for DiskANN, which has a slightly different insertion procedure from HNSW.

- Our algorithm SPatch also requires a hyper-parameter $\alpha$, which determines how many edges to keep in the clique after sparsification. Intuitively, the larger the $\alpha$, the denser the graph. Through out experiments, we observe that choosing $\alpha = 1.2$ except for GIST yields good performances. In

---

**Algorithm 5** Construction via sparsification.

---

1: **procedure** CONSTRUCTONELAYER($P \in (\mathbb{R}^d)^n, m \in [n], r \in \mathbb{R}_+$)
2:     Determine an ordering of the points in $P$, label them as $v_1, \ldots, v_n$
3:     $G \leftarrow (\{v_1\}, \emptyset)$
4:     **for** $i = 2 \to n$ **do**
5:         `candidates` $\leftarrow \{v_1\}$
6:         `visited` $\leftarrow \{v_1\}$
7:         $V \leftarrow V \cup \{v_i\}$
8:         // Densify phase
9:         $E \leftarrow E \cup \{\{v_j, v_i\} : j \in [i-1], w(v_j, v_i) = \exp(-r^2 \cdot \|v_j - v_i\|^2)\}$
10:        // Random walk phase
11:        **while** $|$`candidates`$| > 0$ **do**
12:           Sample $c \in$ `candidates` with probability $\frac{\exp(-r^2 \cdot \|v_i - c\|^2)}{\sum_{u \in \text{candidates}} \exp(-r^2 \cdot \|v_i - u\|^2)}$
13:           `candidates` $\leftarrow$ `candidates` $\setminus \{c\}$
14:           **for** $u \in \text{nbrhood}(c)$ **do**
15:             **if** $u \notin$ `visited` **then**
16:               `visited` $\leftarrow$ `visited` $\cup \{u\}$
17:               `candidates` $\leftarrow$ `candidates` $\cup \{u\}$
18:             **end if**
19:           **end for**
20:        **end while**
21:        // Sparsify phase
22:        `sample` $\leftarrow$ Sample $m$ points from `visited` independently without replacement, with probability of sampling $u$ being $\frac{\exp(-r^2 \cdot \|v_i - u\|^2)}{\sum_{v \in \text{visited}} \exp(-r^2 \cdot \|v_i - v\|^2)}$
23:        $E \leftarrow E \setminus \{\{v_j, u\} : u \notin \text{sample}\}$
24:        **for** $u \in$ `sample` **do**
25:           **if** $\deg(u) > m$ **then**
26:             Sparsify $\text{nbrhood}(u)$ by sampling $m$ edges
27:           **end if**
28:        **end for**
29:     **end for**
30: **end procedure**

---

particular, this consistently holds for `MPNet` and `MiniLM`. For `SIFT`, we could further improve the recall and efficiency by choosing $\alpha = 0.6$. For `GIST` however, one has to choose $\alpha$ to be smaller than 1 to obtain good recall and efficiency. In our experiment, we choose $\alpha = 0.4$. We summarize the choices in Table 3. To choose $\alpha$, we recommend either using $\alpha = 1.2$ or $\alpha = 0.6$.

|  | SIFT | GIST | MPNet | MiniLM |
|---|---|---|---|---|
| $\alpha$ | 0.6 | 0.4 | 1.2 | 1.2 |

Table 3: Choices of hyper-parameter $\alpha$ for different datasets.

**Random softmax walk vs. greedy search.** For this experiment, we need to choose a hyper-parameter $r$ (recall the softmax walk samples for the next visit with probability $\exp(-r^2 \cdot \|q - u\|^2)$). Intuitively, we want to choose $r$ so that the softmax walk samples the nearest neighbor with exponentially higher probability than the second nearest neighbor. This could be achieved by choosing $r$ to be arbitrarily large. However, if $r$ is chosen to be too large, the probability can easily overflow. To resolve this issue, we adapt the following approach for computing $r$: given a collection of points `candidates` to consider (line 7 of Algorithm 4), we compute the empirical average of the distances $\mu = \sum_{u \in \text{candidates}} \frac{\|u - q\|}{|\text{candidates}|}$, then we set $r = 15/\mu$. This scales $r \cdot \|q - u\|$ to a value between 10 and 20, and empirically, we observe that this choice of $r$ can differentiate among the top nearest neighbors and henceforth, give similar recall as the greedy search.

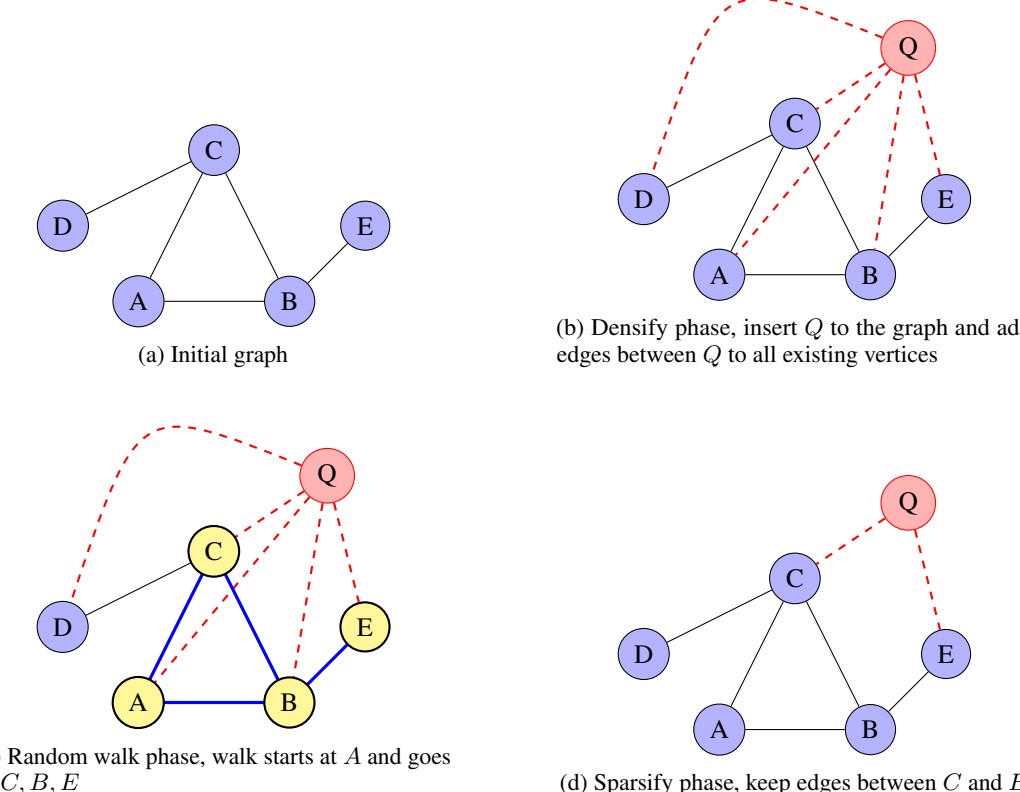

(a) Initial graph

(b) Densify phase, insert $Q$ to the graph and add edges between $Q$ to all existing vertices

(c) Random walk phase, walk starts at $A$ and goes to $C, B, E$

(d) Sparsify phase, keep edges between $C$ and $E$

Figure 3: We construct the graph by first adding edges between $Q$ to all vertices, then perform a random walk to determine the candidate edges to keep, and sparsify them by sampling.

Figure 4 justifies this decision to fix $r\mu = 15$ by demonstrating the behavior of Random Softmax search for different values $\widehat{r} := r\mu$. The left plot shows that as $\widehat{r}$ increases, the softmax converges toward a true maximum, and randomized softmax correspondingly better matches the greedy search algorithm. Thus, $\widehat{r}$ in this regime usually – but far from always – transitions to the current node's neighbor closest to $q$. The right plot shows the impact of $\widehat{r}$ on recall, and in particular its convergence to the performance of the pure greedy algorithm (dashed horizontal lines) for $\widehat{r} \approx 15$. **Deletion time**

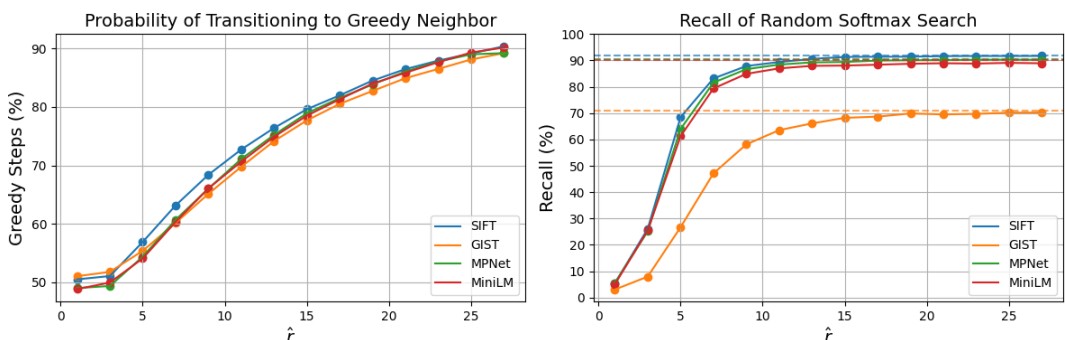

Figure 4: The impact of varying $\widehat{r}$ (i.e. $r\mu$) on transition probabilities and recall. Left: The frequency with which the random softmax algorithm truly transitions to the nearest neighbor (i.e. a greedy step), as a function of $\widehat{r}$. Right: The impact of different choices of $\widehat{r}$ on the recall of the randomized search algorithm. Horizontal line indicates the recall of greedy search algorithm.

**without FreshDiskANN.** As we have observed in the experiment, FreshDiskANN has much slower

deletion time than other methods, in the following figures, we provide deletion time comparison *without* FreshDiskANN.

Figure 5: Top left: `SIFT`, top right: `GIST`, bottom left: `MPNet`, bottom right: `MiniLM`.

### E.2 STEADY STATE SETTING

In this experiment, we consider the steady state setting introduced in Singh et al. (2021), where 10% of the points are deleted from the data structure then inserted back, and queries are measured. We repeat this process 10 times, albeit all points have been deleted from the data structure and reinserted. Similar to the mass deletion experiment, we randomly pick 5,000 query points for `SIFT`, `MPNet`, `MBREAD` and `MiniLM` and 1,000 query points for `GIST`.

Through Figure 6, we could see a similar trend as in the deletion experiment (Section 5), except that the no patching algorithm gives better recall than before. This is because in the steady state setting, the graph is automatically "patched" by re-inserting the same set of points back to the data structure. In contrast to the experimental results in Singh et al. (2021), FreshDiskANN does not perform even as well as no patching, this in part is because the insertion algorithms for DiskANN and HNSW are quite different. Regarding hyper-parameters: for FreshDiskANN, we again choose $\alpha = 1.2$, and for `SPatch`, we summarize it in Table 4.

|  | SIFT | GIST | MPNet | MBREAD | MiniLM |
|---|---|---|---|---|---|
| $\alpha$ | 0.5 | 0.5 | 1.2 | 1.6 | 1.2 |

Table 4: Choices of hyper-parameter $\alpha$ for different datasets.

### E.3 VERTEX COUNTS, EDGE COUNTS, AND MEMORY UTILIZATION

To see the impact of `SPatch` on the graph size and empirical memory utilization, we design an alternative steady-state experiment in an environment with very high vector turnover. Starting with an empty database, we continuously insert into it one vector per (simulated) second from the `MiniLM` dataset over the duration of 12 (simulated) hours. Each vector survives for a number of steps following an exponential distribution with a mean of 2 hours (i.e. has a half-life of $2\ln 2 \approx 1.39$ hours), after which it is deleted. Hence, while the experiment has more insertions than deletions for the first few simulated hours, it eventually enters a steady state at which insertions and deletions occur at roughly the same rate. At the end of the 12 hours, no new vectors are inserted, and the remaining vectors continue to fall out as they expire.

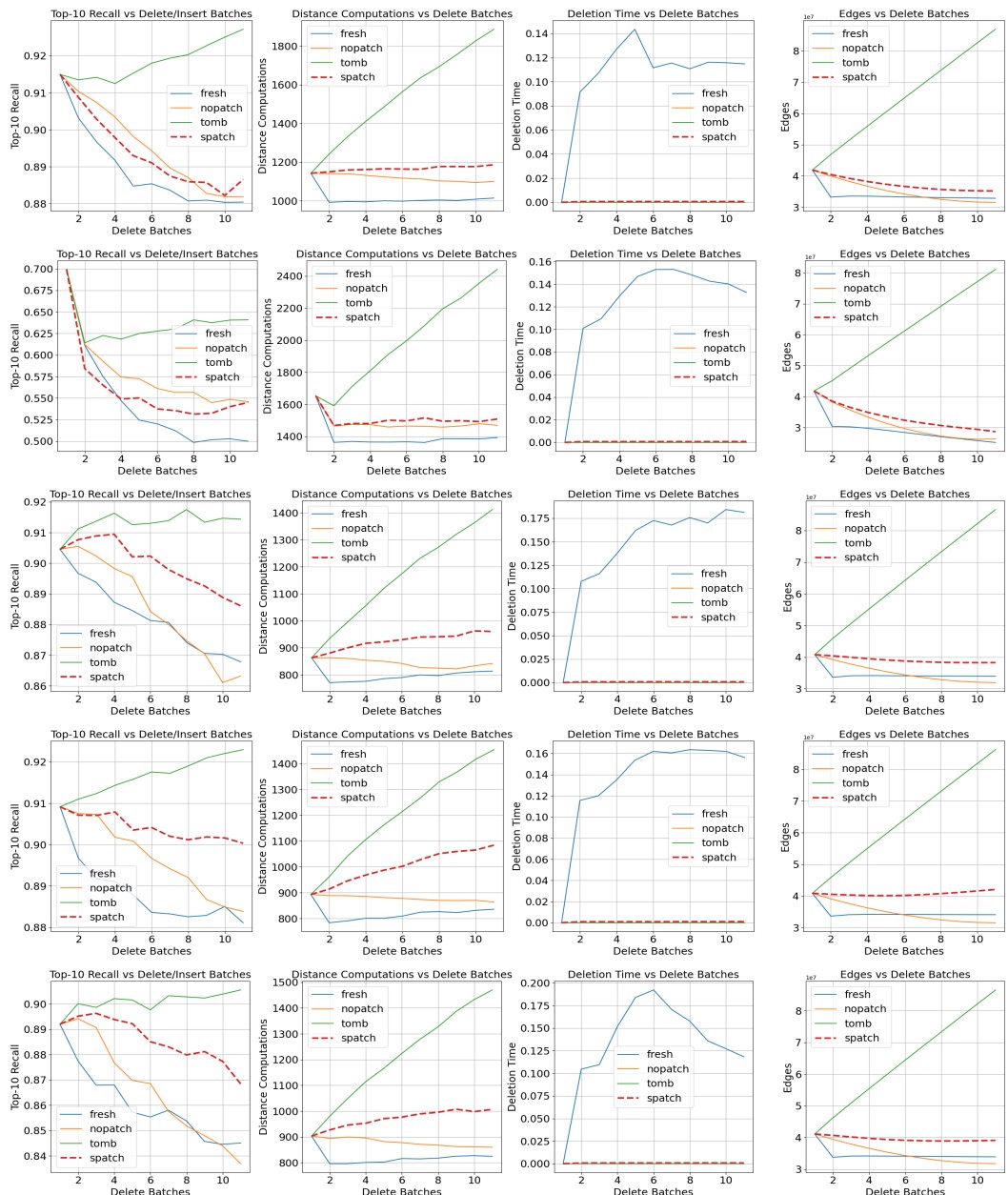

Figure 6: The $5 \times 4$ grid of figures, where the rows are SIFT, GIST, MPNet, MBREAD and MiniLM, and the columns are recall, number of distance computations per query, total deletion time and total deletion time excluding fresh DiskANN. Legends: spatch – our deletion algorithm SPatch, fresh – FreshDiskANN algorithm, tomb – tombstone algorithm, nopatch – no patching algorithm.

We track the vertex counts, edge counts, and memory utilization of both the Tombstone algorithm and of SPatch, and the results are shown in Figure 7. Vertex and edges are counted cumulatively over all levels of the graph. Hence, we continue to see a small rise even in the steady state: the upper layers of the data structure still continue to accumulate them at their prior rate. However, the rate of increase is *far* slower than that of the Tombstone algorithm, which intrinsically grows at a steady rate independent of the number of deletions.

We also see a stark difference in memory utilization (as measured by python's `psutil` package). While the `Tombstone` algorithm is marginally more memory efficient before the effect of deletions begins to kick in (as a result of storing approximately $d + m = 384 + 32 = 416$ words per vector instead of $d + 2m = 448$ due to the bidirected nature of the HNSW graph), soon the vector deletion begins saving large amounts of memory due to the combined effect of the vertices, edges, and $d$-dimensional vectors removed from the data store. Although vertex, edge, and memory all continues to rise with `SPatch`, it does so substantially slower than with `Tombstone`. Finally, after we enter the pure-deletion phase of the experiment 12 hours in, we see no further change in *any* of the plots for `Tombstone` (as expected), but each of the metrics begins its decrease for `SPatch`. While not all of the memory can be directly recovered by the Operating System due to fragmentation or other similar considerations, we do see the memory utilization of `SPatch` begin to dip, and (unlike in `Tombstone`) much of the unrecovered memory is ready for reuse by the algorithm should it face more insertions in the future.

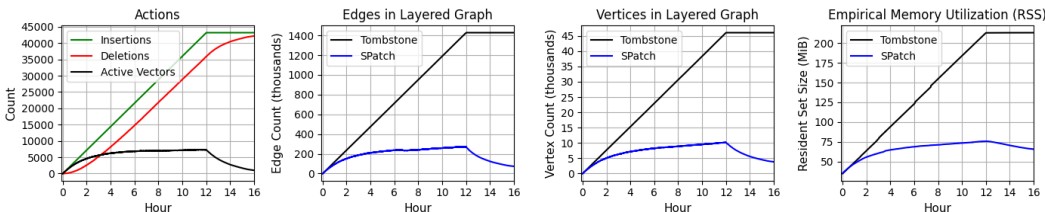

Figure 7: Vertex counts, edge counts, and empirical memory utilization as measured by the experiment described in Section E.3.

### E.4 SOFTMAX WALK AND GREEDY SEARCH: A MORE DETAILED COMPARISON

In this section, we give a more detailed comparison between softmax walk and greedy search. For the first set of plots, we plot beam size vs recall@10, and for the second set of plots, we plot the total number of distance computations vs recall@10 in Figure 8. We can see that fixing the beam size, softmax walk gives slightly lower recall, but the second row shows that fixing the number of distance computations, softmax walk is very close to greedy search.

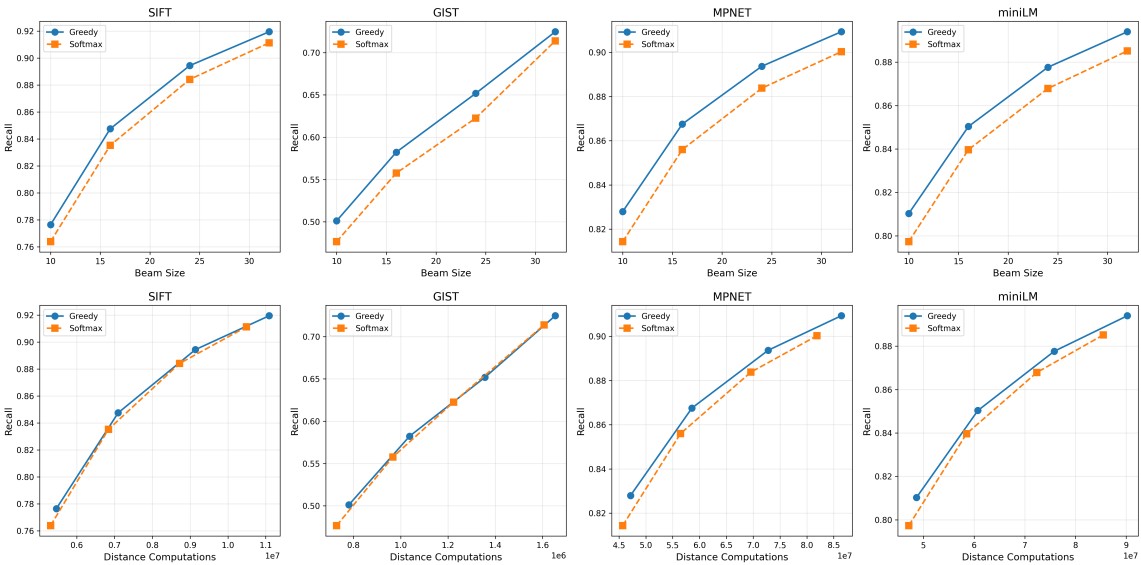

Figure 8: From left to right: `SIFT`, `GIST`, `MPNet` and `MiniLM`. Top row: beam size vs recall@10, bottom row: total number of distance computations vs recall@10.

## E.5 EXPERIMENTS ON DISKANN

In this section, we provide preliminary experiments on DiskANN, specifically we test `SPatch`, FreshDiskANN, tombstone, local and no patch algorithm. We test it on a subsampled 100k points from `SIFT` dataset, each deletion batch, we delete 800 points then perform a query. Compared to the experiments on HNSW, FreshDiskANN has improved performance in terms of recall, however, we note the deletion time of FreshDiskANN is still very large.

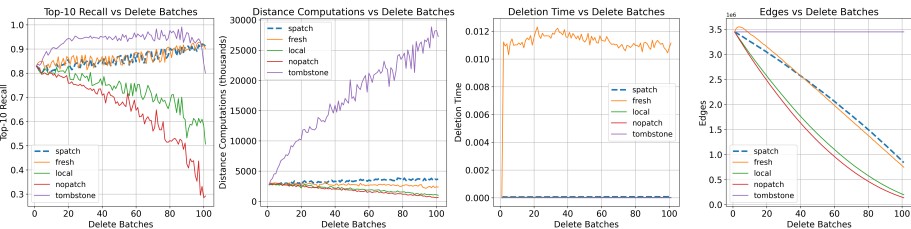

Figure 9: Experiments for DiskANN on 10% of points from `SIFT`.

## E.6 EXPERIMENTS ON DYNAMIC EXPLORATION GRAPHS (DEG)

We also provide preliminary comparisons to the deletion strategy used in the Dynamic Exploration Graph (DEG) data structure of Hezel et al. (2025). In DEG, the deletion of a vertex $v$ is patched by building an independent BFS trees from *each* neighbor of $v$ and adding an edge between two of these neighbors when their corresponding trees collide, until the total degree on the neighborhood is restored. Like in the previous section, we subsample 100k points from `SIFT` and monitor (i) recall, (ii) distance computations per query, (iii) deletion time, and (iv) graph size after repeatedly deleting 10K points.

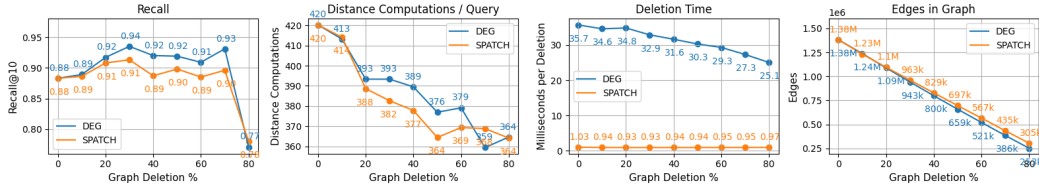

Figure 10: Experiments for DEG on 10% of points from `SIFT`.

The results are shown in Figure 10. Note that DEG's deletion time is much slower than our method, SPATCH. In the cost/accuracy trade-off, SPatch and DEG perform very similarly, with DEG achieving slightly higher recall (within 2-3%) with a slightly higher number of distance computations, rendering their cost/accuracy tradeoff comparable. The crucial difference is deletion time: **the deletion time of DEG is up to 35× larger than SPatch**, reflecting the inherent cost of DEG's global rebuild strategy compared to SPatch local updates. This gap makes DEG impractical when deletions are frequent or substantial, which is precisely the regime our method targets. While DEG achieves slightly higher recall than SPatch, its much slower deletion time makes it impractical for use in scenarios where deletions are plentiful.

## LLM DISCLOSURE

This work does not use LLM to facilitate writing.

