# OpenReview forum: "Graph-based Nearest Neighbors with Dynamic Updates via Random Walks"
_ICLR.cc/2026/Conference — ICLR 2026 Poster_

### Official Review · Reviewer_G68m · 2025-10-30

**Soundness:** 3
**Presentation:** 3
**Contribution:** 3
**Rating:** 8
**Confidence:** 3

**Summary:**

This paper focuses on approximate nearest neighbor search problem. Based on the existing Hierarchical Navigable Small World (HNSW) data structure, this paper proposes SPatch, a novel random walk-based approach to support deletion of existing data. The experimental results on various datasets in massive deletion settings show the proposed method provides better tradeoff between query latency, recall, deletion time and memory usage.

**Strengths:**

1. This paper proposes a novel deletion algorithm for HNSW, a widely adopted data structure in vector search.
2. This paper provides both theoretical and practical analyses on the proposed algorithm.
3. The proposed algorithm achieves good tradeoff between four key metrics.

**Weaknesses:**

1. As mentioned in the paper, deletion procedure involves more complicated operations. I wonder if it is possible to provide time and space complexity analyses on the deletion algorithms.
2. The experimental setting of this paper seems to be different from previous works, such as FreshDiskANN and SPFresh. They simulate the scenario for both insertion and deletion, which are more closed to the real world.

**Questions:**

See weaknesses above.

---

> ### Author Response · Authors · 2025-11-22
>
> Thank you for your very insightful comments. We are happy to address your comments.
>
> * Regarding the runtime of our deletion algorithm: let $p$ be the point we want to delete, if we let $N_{\rm in}(p)$ denote the in-neighbors to $p$ and $N_{\rm out}(p)$ denote the out-neighbors of $p$, then the runtime of deletion is $O(|N_{\rm in}(p)|\cdot |N_{\rm out}(p)|\cdot d\cdot \log(|N_{\rm in}(p)|\cdot |N_{\rm out}|))$, as our algorithm computes pairwise distances between the in-neighbors and out-neighbors of $p$, then updates weights accordingly. The sampling or pick top-$t$ incurs at most an additional log factor.  We will add a runtime discussion to the paper in Appendix D.3.
>
> * Regarding experiments on both insertion and deletion: in Appendix E, we perform additional experiments on the steady state setting studied in [1], where we iteratively delete 5% of points, then insert them back, and test recall, number of distance computations and deletion time.
>
> Reference
>
> [1] FreshDiskANN: A Fast and Accurate Graph-Based ANN Index for Streaming Similarity Search. Singh, Subramanya, Krishnaswamy and Simhadri. 2021.

---

### Official Review · Reviewer_Z2rF · 2025-10-30

**Soundness:** 2
**Presentation:** 2
**Contribution:** 2
**Rating:** 2
**Confidence:** 4

**Summary:**

The paper presents a theoretical framework for graph-based approximate nearest neighbor (ANN) search grounded in random walk analysis. Using this framework, the authors analyze a randomized deletion strategy that maintains hitting time statistics comparable to those of the original graph. They further leverage the theoretical insights to develop a deterministic deletion algorithm and experimentally compare it with other deletion algorithms.

**Strengths:**

S1: Deletion is important operation for dynamic similarity graphs, and any related research is welcome.

S2: The theoretical analysis of deletion in similarity graphs seems reasonable.

**Weaknesses:**

W1. A highly related work, DEG [a], is missing. In [a], the authors also proposed a similarity graph capable of handling all updates, including deletions. It is recommended that the proposed approach, Spatch, be compared with [a] thoroughly in terms of both design and performance.

W2. The authors use rebuild as the baseline. However, a recently proposed approach [b] can rebuild the HNSW index much faster without impairing search performance. The authors are recommended to implement rebuild according to the method in [b] for the performance comparison.

W3. Some explanations are confusing. On line 425, the authors claim that, compared with rebuild, the query speed of Spatch is much faster and memory usage decreases as more points are deleted. However, from the results in Fig. 2, in many cases Spatch requires more distance computations than rebuild. How should the statement that Spatch is faster be interpreted? Additionally, where are the experimental results on memory usage?

W4. (Minor) I feel this paper is more suited to conferences on databases or data mining, where handling updates is a primary focus.

[a] Dynamic exploration graph: a novel approach for efficient nearest neighbor search in evolving multimedia datasets. International Conference on Multimedia Modeling 2025.

[b] Revisiting the Index Construction of Proximity Graph-Based Approximate Nearest Neighbor Search. VLDB 2024

**Questions:**

Please see the weaknesses and address the concerns mentioned there.

---

> ### Author Response · Authors · 2025-11-22
>
> Thank you for your valuable comments and we are happy to address your concerns.
>
> * Regarding comparison to DEG [1]. The deletion algorithm proposed by DEG tries to maintain the reachability of the navigation after deletion by carefully examining the connectivity of $N(p)$, where $p$ is the deleted point. For each $v \in N(p)$, it starts by initializing the set of reachable vertices as themselves, then using BFS to update the set of reachable vertices. When there is more than one reachable component, it iteratively connects two components by picking a pair of vertices that are closest and adding the edge.  It then repeats this process until there is only one component. Then for vertices within a component, an edge is added to its nearest neighbor within the component. Finally, handling edge cases by using 2-hop neighborhoods, and optionally optimize edges. Compared to our algorithm, DEG emphasizes more on ensuring connectivity after deletion via BFS and 2-hop correction, while our algorithm simply reroutes among $N(p)$ via a novel sparsification scheme. We are currently performing an empirical comparison with the DEG algorithm, and will update once the experiments complete.
>
> * Regarding comparison to faster rebuild [2]. Thank you for pointing out the reference. [2] proposes a faster construction algorithm for HNSW, which would potentially improve the rebuild time of our baseline. Their experiments achieve up to 5.6x speedup over vanilla construction. Even with this speedup, rebuilding the entire graph is still much slower than making a local change to the graph. In the following table, we summarize the total deletion time, on average, for 8k points in seconds, for the datasets we experimented on. For faster rebuild [2], we count the rebuild time as the deletion time.
>
> |                | SIFT  | GIST  | MPNet | MiniLM |
> |----------------|-------|-------|-------|--------|
> | SPatch (ours)        | 8.15  | 5.26  | 5.83  | 6.24   |
> | Faster Rebuild [2] | 27.86 | 97.14 | 37.86 | 42.32  |
>
> Even with the faster rebuild of [2] and their validated speedup, rebuilding is still significantly slower. Moreover, it does not provide the correct “semantics” for a deletion algorithm in the sense that if we query when a point has been removed but the rebuilding has not been triggered, the query would potentially return the deleted point, hence extra care is needed to ensure that these deleted points not to be returned.
>
> * Regarding the claim on Line 425: Thank you for pointing it out, the query speed of SPatch is indeed not faster than periodic rebuild. It is faster than tombstone and has decreased memory usage. It has much faster deletion time than both FreshDiskANN and periodic rebuild. We revised this part in the manuscript.
>
> * Regarding memory experiments: Experiments on memory appear in Appendix E.3, no space in the main body. In short, we simulate a steady-state setting where vectors are inserted into the data structure, then survive for an amount of time that follows an exponential distribution. We compare Spatch with tombstone, and track the vertex and edge counts, and memory utilization by Python’s psutil. One can see a clear contrast in all these memory measurements between SPatch and tombstone, as tombstone keeps increasing the memory and flattens when no points are inserted, and SPatch starts to reduce the memory once points have been deleted.
>
> References
>
> [1] Dynamic exploration graph: a novel approach for efficient nearest neighbor search in evolving multimedia datasets. International Conference on Multimedia Modeling 2025.
>
> [2] Revisiting the Index Construction of Proximity Graph-Based Approximate Nearest Neighbor Search. VLDB 2024.

---

> > ### Comment · Reviewer_Z2rF · 2025-11-28
> >
> > Thank you for your explanation. Regarding W3, the proposed method is actually slower than rebuild and achieves lower recall than rebuild. Considering the absence of a comparison with DEG, I am concerned about whether the proposed method is truly useful. For the comparison with other baselines, I suggested that the authors plot the QPS–recall curve, because with the current figures it is difficult to know which method performs better. For example, Tombstone shows higher recall than the proposed method but requires more distance computations, so the comparison is not straightforward.

---

> > > ### Author Response · Authors · 2025-12-02
> > > **Update on DEG experiments & Response on the comparison vs rebuild**
> > >
> > > In response to the reviewer's request, we have added a preliminary comparison to DEG, a global rebuild algorithm. DEG's deletion time is up to 35x slower than our method, SPATCH. The full experiment has been added to Appendix E.6.
> > >
> > > **Experiment:** We sample 100k points at random from SIFT and then iteratively delete 10000 points until 80% of the vectors are removed. As in the main paper, we measure recall@10 (accuracy) vs. the number of distance computations (cost).
> > >
> > > **Results:** In the cost/accuracy trade-off, SPatch and DEG perform very similarly, with DEG achieving slightly higher recall (within 2-3\%) with a slightly higher number of distance computations, rendering their cost/accuracy tradeoff comparable.
> > > The crucial difference is *deletion time*: the deletion time of DEG is up to 35× larger than that of SPatch, reflecting the inherent cost of DEG’s global rebuild strategy compared to SPatch local updates. This gap makes DEG impractical when deletions are frequent or substantial, which is precisely the regime our method targets.
> > >
> > > **General remark:** The reviewer’s observation that a full rebuild can yield slightly higher recall at comparable query cost is consistent with our findings. We do not claim advantages for SPatch in purely static settings where the index is built once and never updated. Our contribution is a method that offers competitive query-time accuracy/cost trade-off while supporting efficient deletions at scale in regimes where repeated global rebuilds quickly become prohibitively expensive.

---

### Official Review · Reviewer_rt4u · 2025-11-01

**Soundness:** 3
**Presentation:** 3
**Contribution:** 3
**Rating:** 6
**Confidence:** 5

**Summary:**

The paper studies the problem of deletions of base points in the context of Approximate Nearest Neighbor (ANN) search. It proposes a theoretically grounded procedure that efficiently updates the ANN data structure upon deletions. The proposed approach achieves better recall, along with good query performance, efficient deletion update time, and reasonable memory usage.

Conclusion: Although the results presented in the paper are interesting, there are lot of missing pieces. I am leaning in the middle, I don't mind either a accept or reject for this paper.

**Strengths:**

I see two main strengths in this paper:

The first is the novel perspective of interpreting the greedy walk through the lens of a softmax walk. This conceptual shift provides valuable intuition and a unifying view of the search process.

Building on this softmax interpretation, the paper proposes a graph sparsification approach, where handling deletions efficiently naturally corresponds to sparsifying a complete weighted graph.

These viewpoints together lead to a practical and effective deletion procedure that performs well empirically.

**Weaknesses:**

1) I feel there is a little disconnect between theory and its relation to guarantees achieved by the ANN procedure. For instance, I would have loved to see a theorem statement of the kind, for any query q, if greedy walk on G achieves alpha approx or recall, then softmax walk also achieves same approx or recall with high prob. Also, greedy walk or softmax walk on sparsified G' also achieves similar approximation factors.

2) Most of the theorem statements are simple. I am not so impressed with the proofs except this viewpoint of seeing greedy walk through the lens of softmax walk.

3) Also the softmax walk was defined with respect to query point q, but SPatch algorithm uses distances between base points to sparsify the graph. There was no explanation given on why this is okay. I am aware that queries are not given to us, somehow an explanation is needed on why sparsifying graph based on distances between base points is okay. When inserting a base point v into the graph, usually these graphs are constructed assuming query is v itself, and this assumption is used to decide the incoming and outgoing edges of v. It would have been nice to see some explanation of SPatch algorithm with respect to this assumption of viewing base points as query points.

**Questions:**

1) You keep using the word "twin formulation", what does twin refers to?
2) Theorem 4.3 implies a bound in Theorem 4.5, can you please also add some comments on what the bounds in Theorem 4.5 would look like if you had approximations in terms of spectral norms than the Forbenius norm in Theorem 4.3. Also, talk about the the sparsity bounds for the spectral norm from prior work.
3)  Please explain beam search version of softmax clearly in your paper. I am assuming that, during search, at each hop, suppose you have your current list L of size t and you expand a node v in L. Then you look at L \cup N(v) and sample top t edges based on their softmax probabilities? Also, what is the stopping criterion for the softmax search, that is, when will the search stop?
4) In Table 2, though there is a recall drop, but the number of distance comparisons done by softmax walk is also lower. To have better clarity, for each dataset, can you create a plot with recall@10 numbers (y-axis) for different beam sizes (x-axis)?
5) Also in Figure 2, can you generate different plots, where x-axis is distance comparisons and y-axis is recall@10? The number of distance comparisons can vary by varying the beam size.
6) Clearly mention in the introduction itself when is your procedure good by looking at Figure 2. For instance, your procedure is good when deletion batch size is higher because of blah reason. It would be nice to add some conclusions drawn from experiments in the introduction itself.
7) Handle weakness 3.

---

> ### Author Response · Authors · 2025-11-22
>
> Thank you for your very insightful review. We are happy to address your concerns and questions.
>
> * Regarding obtaining an $\alpha$-approximation guarantee with softmax walk: Thank you for raising this point. Note that proving even the vanilla greedy search algorithm works (i.e., achieving high recall, or providing certain approximation guarantees) is highly nontrivial, and to the best of our knowledge, all known such results either require extra structural assumptions on the graphs or query points [1] or make simplifying assumptions [2]. On the other hand, our analysis makes minimal assumptions (the graph is undirected). It is possible that by imposing stronger assumptions, one can prove that a softmax walk indeed gives approximation or recall guarantees.  It’s a good direction for future work.
>
>     As softmax walk is defined with respect to the query point $q$, how to justify its use when using SPatch: Recall the process of constructing the graph: points are iteratively inserted, when a new point $v$ is inserted to the graph, we are essentially treating it as a query point with slightly different parameters for beam search, and adding edges accordingly. Using this perspective, we can record the edge weights whenever a new point $v$ is inserted and becomes a base point, therefore when performing the SPatch deletion, we use weighted edges to perform sampling. We have added a paragraph discussing this view point in the revision in Appendix D.4.
>
> * Regarding the use of “twin” formulation: We use the word “twin” to refer to two main contributions of our work (1) a theoretical framework that uses randomness in softmax walk and Frobenius norm sparsification to analyze graph-based ANN and inspire new algorithm designs for these models. (2) a practical realization of these ideas that integrate them into existing graph-based ANN by replacing the softmax walk with the greedy search, and replacing the weighted sampling by deterministic top-$t$ selection. The bridge between this twin formulation is also experiments showing that in practice, the performance (in terms of recall) of these two methods are very similar, while the deterministic version is more efficient.

---

> > ### Author Response · Authors · 2025-11-22
> >
> > * Regarding spectral approximation instead of Frobenius norm: If we obtain spectral approximation of the graph via leverage score sampling [3] or a more sophisticated approach [4], then we will obtain a graph with Laplacian matrix $\widetilde L$ such that $(1-\epsilon) L\preceq \widetilde L \preceq (1+\epsilon)L$ where $A\preceq B$ means that $B-A$ is PSD. The key observation is that the hitting time can be written as a solution to a Laplacian system, in particular, to compute the hitting time vector to a vertex $v$, let $L_{\setminus v}$ denote the Laplacian matrix removing the row with respect to $v$ and $d_{\setminus v}$ denote the degree vector removing the degree of $v$, then the hitting time vector $h_{\setminus v}$ is the solution to $L_{\setminus v} h_{\setminus v}=d_{\setminus v}$. Given $\widetilde L$, we solve instead $\widetilde L_{\setminus v} \widetilde h_{\setminus v}=\widetilde d_{\setminus v}$, where $\widetilde d$ is the degree vector of $\widetilde L$. One can then see that $\widetilde h_{\setminus v}$ approximates $h_{\setminus v}$ in the $L$-norm, as follows: note that both $L_{\setminus v}$ and $\widetilde L_{\setminus v}$ are full rank, a standard fact from spectral graph theory, we can write these solutions in matrix inverses: $h_{\setminus v}=L_{\setminus v}^{-1} d_{\setminus v}$ and $\widetilde h_{\setminus v}=\widetilde L_{\setminus v}^{-1} \widetilde d_{\setminus v}$. To simplify the notation, we ignore the subscript $\setminus v$, then we can bound the error in $L$-norm, $\\|h-\widetilde h\\|\_{L}$ as $\\| L^{-1}d- \widetilde L^{-1}d\\|\_{L}+\\| \widetilde L^{-1}(d-\widetilde d)\\|\_{L}$. For the first term, we expand the square: $d^\top (L^{-1}-\widetilde L^{-1}) L (L^{-1}-\widetilde L^{-1})d=d^\top L^{-1}L^{1/2} (L^{1/2}(L^{-1}-\widetilde L^{-1}) L(L^{-1}-\widetilde L^{-1})) L^{1/2}L^{-1} d\leq \\|L^{1/2}(L^{-1}-\widetilde L^{-1})L^{1/2} \\|^2\cdot \\|L^{-1}d\\|\_L^2$, as $\widetilde L$ is a spectral approximation of $L$, we know that $\\|  L^{1/2}(L^{-1}-\widetilde L^{-1})L^{1/2} \\| = \\| I-L^{1/2}\widetilde L^{-1}L^{1/2}\\|\leq \epsilon$, thus we obtain an upper bound $\epsilon\cdot \\|L^{-1}d\\|\_{L}=\epsilon\cdot \\|h\\|\_L$ for the first term. For the second term, we note that all degrees of $\widetilde L$ is a $(1\pm\epsilon)$ approximation to the degrees of $L$ by testing with vectors $e_i$’s. Further, we note that the Laplacian matrix removing the $v$-th row/column is an M-matrix, hence its inverse has all positive entries. Let $\delta=d-\widetilde d$, then $\\|\widetilde L^{-1} \delta\\|\_L^2=\delta^\top \widetilde L^{-1} L \widetilde L^{-1} \delta \leq (1+\epsilon) \delta^\top \widetilde L^{-1} \delta \leq (1+\epsilon)^2 \delta^\top L^{-1} \delta$, we then write the last quadratic form entrywise: $\delta^\top L^{-1} \delta=\sum\_{i,j} L^{-1}\_{i,j} \delta_i \delta_j \leq \sum\_{i,j}L^{-1}\_{i,j} |\delta_i| |\delta_j| \leq \sum\_{i,j} L^{-1}\_{i,j} \epsilon^2 d_i d_j = \epsilon^2 d^\top L^{-1} d=\epsilon^2 \\|h\\|\_L^2$, combining we have shown that the second term is at most $O(\epsilon) \\|h\\|\_{L}$. Finally, by triangle inequality, we prove that $\\|h-\widetilde h\\|\_L\leq O(\epsilon)\cdot \\|h\\|\_{L}$.
> >
> >
> >     For the sparsity bound of spectral sparsification, [3] gives $O(\epsilon^{-2}n\log n)$ with leverage score sampling, and [4] gives $O(\epsilon^{-2}n)$ via more sophisticated approach to remove the $\log n$ factor. The algorithm of [4] has been improved in later works [5, 6, 7, 8].
> >
> > * Regarding beam search version of softmax walk: Yes, the beam search version is precisely what you described: Given a list of $t$ points currently maintained, we expand the neighborhood at $v$ into $L$, then sample $t$ edges according to edge weights. In our experiment, we replace this procedure with the deterministic version: instead of sampling $t$ edges, we simply keep the top-$t$ heaviest edges. The stop criterion is similar to the standard greedy search of HNSW: given the current point $v$, we sample a neighbor of $v$ according to the edge weight. If the distance between this sampled neighbor is further than any points we have maintained in the candidate list, we stop. You can find Algorithm 4 in the appendix (no space in the main body).
> >
> > * Regarding plots on beam size vs recall for softmax walk and number of distance computations vs recall for SPatch: We are currently working on these experiments and generating plots.  Once the experiments complete, the paper will be updated.
> >
> > * Regarding when our deletion procedure is good: Our deletion algorithm is particularly strong in the decremental setting, where large batches of points are being removed from the graph frequently, it maintains high recall and low memory footprint. We added this discussion in the introduction.

---

> > > ### Author Response · Authors · 2025-11-22
> > >
> > > References
> > >
> > > [1] Worst-case performance of popular approximate nearest neighbor search implementations: guarantees and limitations. Indyk and Xu. NeurIPS 2023.
> > >
> > > [2] Navigable graphs for high-dimensional nearest neighbor search: Constructions and limits. Diwan, Gou, Musco, Musco and Suel. NeurIPS 2024.
> > >
> > > [3] Graph sparsification by effective resistances. Spielman and Srivastava. STOC 2008.
> > >
> > > [4] Twice Ramanujan sparsifier. Batson, Spielman and Srivastava. STOC 2009.
> > >
> > > [5] A matrix hyperbolic cosine algorithm and applications. Zouzias. ICALP 2012.
> > >
> > > [6] Spectral Sparsification and Regret Minimization Beyond Matrix Multiplicative Updates. Allen-Zhu, Liao and Orecchia. STOC 2015.
> > >
> > > [7] Constructing Linear-Sized Spectral Sparsification in Almost-Linear Time. Lee and Sun. FOCS 2015.
> > >
> > > [8] An SDP-Based Algorithm for Linear-Sized Spectral Sparsification. Lee and Sun. STOC 2017.

---

> > > > ### Author Response · Authors · 2025-11-27
> > > > **Update on requested comparison: beam size vs recall@10 and distance computations vs recall@10**
> > > >
> > > > We would like to update that we have added the requested plots in Appendix E.4 in the revised manuscript.

---

### Official Review · Reviewer_j4UT · 2025-11-08

**Soundness:** 3
**Presentation:** 2
**Contribution:** 3
**Rating:** 4
**Confidence:** 3

**Summary:**

### Paper Summary

This paper addresses the critical update problem in graph-based ANN search, focusing on efficient deletion. The core contribution is a new deletion algorithm, SPatch, derived from random walk theory. The problem of removing a node is reformulated as preserving the random walk transition probabilities among the deleted node’s neighbors. Extensive experiments show that SPatch achieves a well-balanced trade-off across four key metrics: high recall, fast query speed, low deletion time, and memory efficiency.

**Strengths:**

### Paper Strengths

S1. Efficient, high-recall dynamic deletion is one of the most significant unsolved challenges for graph-based ANN indexes in production. The paper’s motivation is strong and highly relevant.

S2. The idea of modeling a graph’s greedy search as a “softmax walk” is clever, bridging a heuristic algorithm with formal random walk theory.

S3. The paper includes comparisons against mainstream baselines and demonstrates competitive performance across recall, query speed, deletion time, and memory usage.

**Weaknesses:**

### Paper Weaknesses

W1. The paper’s theoretical foundation is built on undirected graphs, whereas mainstream implementations use directed graphs, leaving a gap that the paper does not justify. (See D1, D2)

W2. The algorithm does not clearly describe how it handles in-neighbors of a deleted node. (See D3)

W3. The algorithm is only validated on HNSW, and its robustness on other graph indexes (e.g., NSG, DiskANN) is unknown. (See D4)

W4. The algorithm section includes excessive theoretical formulas, making it difficult to follow. (See D5)

### Detailed Comments

D1. The theoretical foundation (including Laplacians, spectral graph theory, hitting times, and the star–mesh transform) is based on undirected graphs. However, most high-performance graph implementations employ directed graphs. The paper does not explain how the undirected theoretical framework applies to a directed implementation, which raises questions about its validity.

D2. Following D1, the paper is ambiguous about whether the HNSW used in Section 5 is directed or undirected, which is crucial for correctly interpreting the results. For example, Figure 2 reports Deletion Time values near zero for SPatch, Local, and NoPatch, which is implausible for realistic directed graphs. This inconsistency suggests the experiments might have been conducted on an undirected structure, contradicting common high-performance settings.

D3. The proposed method does not specify how in-neighbors of the deleted node are handled. Algorithm 1 and Section 5.1 only describe patching out-neighbors, which could leave dangling in-edges and harm graph navigability. Prior research has shown that proper in-neighbor management is essential for maintaining both performance and structural integrity [1]. This point should be clearly addressed.

D4. The algorithm is evaluated only on HNSW. Its effectiveness and generality on other graph-based indexes (e.g., NSG, Vamana, or $\tau$-MNG) remain to be demonstrated.

D5. The algorithm description contains extensive theory and definitions but lacks clear procedural steps, illustrative figures, and simple examples. This makes it difficult for readers to grasp the main idea and follow the technical flow efficiently.

**Questions:**

Please refer to the weakness and detailed comments.

---

> ### Author Response · Authors · 2025-11-22
>
> Thank you for your insightful comments and we are happy to address your comments.
>
> * **Our theory is built on undirected graphs, while mainstream graph-based ANN uses directed graphs**: Thank you for pointing this out. Indeed, our theory is for undirected graphs while in practice, graphs are directed. Note that theoretical studies for graph-based ANN and HNSW are very limited due to the greedy nature of the search procedure on a directed graph. Existing theorems generally make strong assumptions on the graph construction [1] or simplify the model [2]. We instead assume the graph is undirected, a simple and general assumption and believe that if we cannot prove anything in the undirected setting, it will be even more challenging to prove results in the directed setting. Moreover, for undirected graphs, there’s a larger toolkit of graph algorithms one could attempt to integrate with graph-based ANN, making it an appealing model to study. This work can be viewed as a starting point for a more comprehensive theoretical study of graph-based ANN.  The directed setting is a great direction for future work.
>
> * **The algorithm does not describe how to handle in-neighbors of the deleted point, it’s unclear it will work on directed graphs**: Our practical algorithm indeed handles in-neighbors on a directed graph and we have modified Algorithm 1 to reflect that. Our experiment is performed on standard directed HNSW. Specifically, let $p$ be the point to delete and $N_{\rm in}(p)$, $N_{\rm out}(p)$ denote its in- and out-neighbors respectively, then our algorithm computes the distances between $N_{\rm in}(p)$ and $N_{\rm out}(p)$, compute their weights and updated weights $w’$ respectively, then we add directed edges from $N_{\rm in}(p)$ to $N_{\rm out}(p)$ by choosing the top-$t$ heaviest edges to add. Philosophically, our algorithm was inspired by the theory of random walks on undirected graphs, and empirically, we verify that the proposed algorithm indeed works well on directed, graph-based ANN. We then try to understand what type of properties this algorithm preserves, and we hope this  guides future theoretical studies and algorithm design on graph-based ANN.
>
> * **Deletion time for SPatch, Local and NoPatch is close to 0, suggesting the experiment is conducted on undirected graphs**: We would like to clarify that our experiments are conducted on standard *directed* graphs, in particular, we construct the HNSW index using FAISS, and then turn it into a networkx directed graph data structure. The reason the deletion time of those methods is close to 0 is because FreshDiskANN has a much larger deletion time than all other methods. Zooming in, one can see that NoPatch is much faster than both SPatch and Local, while the latter two have comparable deletion time. They are all much faster than FreshDiskANN. We added a figure to the appendix (out of space in the main body).
>
> * **Algorithm description is very theory heavy and lacks procedural steps**: We significantly revised the algorithm with clear procedural steps and how to practically implement it in directed HNSW. We hope this clarifies the confusion and improves the readability of the algorithm. We also significantly updated Figure 1 to better illustrate how our algorithm functions in the directed setting.
>
> * **Algorithm is only tested on HNSW, not on other graph-based ANN**: We  started  testing our algorithm on DiskANN, and will update the results once they are finished.
>
> References:
>
> [1] Worst-case performance of popular approximate nearest neighbor search implementations: guarantees and limitations. Indyk and Xu. NeurIPS 2023.
>
> [2] Navigable graphs for high-dimensional nearest neighbor search: Constructions and limits. Diwan, Gou, Musco, Musco and Suel. NeurIPS 2024.

---

> > ### Author Response · Authors · 2025-11-29
> > **Update on the experiments on DiskANN**
> >
> > We would like to update that we have performed experiments on DiskANN using a randomly sampled 10% of points from the SIFT dataset and the figures have been added to Appendix E.5. While most of the trends are similar to HNSW, we note an improved recall performance of FreshDiskANN, as the algorithm is designed for DiskANN. However, the recall is still similar to SPatch, meanwhile the deletion time of FreshDiskANN is much slower than SPatch. We refer to Appendix E.5 for more details.

---

### Meta-Review · Area_Chair_QDTM · 2025-12-28

**Summary:**

HNSW is the most widely-used graph-based algorithm for approximate nearest neighbor search (ANN), however, efficiently deleting nodes from the graph without incurring significant computational overhead has long remained a challenge. This paper introduces a new theoretical framework for graph-based ANN grounded in random walks, and leverages this framework to design a deletion method that largely removes deletion-related overhead. The proposed method is demonstrated to achieve better tradeoffs between latency, recall, deletion time, and memory usage.

Reviewers agreed that the paper tackles an important and practical problem of deleting nodes in HNSW (G68m, Z2rF, j4UT), the paper provides both theoretical and practical analysis (G68m, Z2rF, rt4u, j4UT), and the proposed algorithm achieves good tradeoffs between four key metrics as claimed by authors (G68m, j4UT).

Concerns
- Missing time and space complexity analyses for deletion algorithms (G68m) – Provided in author responses
Exp setting is inconsistent with prior work such as FreshDiskANN and SPFresh (G68m) – Provided as Appendix E.
- A highly relevant work, DEG, is missing (Z2rF) – Authors clarified the differences and indicated that the discussion will be included in the revision.
- The baseline can be more efficient, with a more recent, fast implementation (Z2rF) – Authors clarified that even with a more optimized baseline, its gains remain smaller than those achieved by the proposed method, and thus would not change the paper’s conclusions.
- Disconnect between theory and the guarantees provided by the method (rt4u) / Theory statements are straightforward and need further explanation (rt4u, j4UT): Further discussed by the authors.
- Assumption of undirected graphs, whereas most ANN graphs are directed (j4UT): The authors explained that this assumption is unavoidable for their analysis and noted that prior work often makes even stronger assumptions.
- Method appears limited to HNSW (j4UT): The authors added experiments on DiskANN. From the AC’s perspective, HNSW is widely regarded as a state-of-the-art method and has clear advantages over DiskANN in certain regimes (e.g., low-latency settings), so stronger gains on HNSW are acceptable.

**Reviewer Concerns:**

Two reviewers (rt4u, G68m) already assigned high ratings. For the other two reviewers (j4UT, Z2rF), I believe the concerns regarding missing related work, the assumption of undirected graphs, and being limited to HNSW are likely to be addressed. Whether they would have been satisfied with responses related to clarify on theoretical analysis is difficult to judge.

**Reviewer Scores:**

rt4u: 6->6 (unchanged)
G68m: 8->8 (unchanged)
j4UT: 4->6 (most concerns are likely to have been addressed)
Z2rF: 2->4 (concerns about missing related work and weak baseline are likely to be addressed, while reviewer may still have concerns related to clarify in theoretical explanation and venue mismatch)

---

### Decision · Program_Chairs · 2026-01-26

Accept (Poster)